# The Ca²⁺ influx through the mammalian skeletal muscle dihydropyridine receptor is irrelevant for muscle performance

Anamika Dayal[1], Kai Schrötter[1], Yuan Pan[2], Karl Föhr[3], Werner Melzer[2] & Manfred Grabner[1]

Skeletal muscle excitation–contraction (EC) coupling is initiated by sarcolemmal depolarization, which is translated into a conformational change of the dihydropyridine receptor (DHPR), which in turn activates sarcoplasmic reticulum (SR) $Ca^{2+}$ release to trigger muscle contraction. During EC coupling, the mammalian DHPR embraces functional duality, as voltage sensor and L-type $Ca^{2+}$ channel. Although its unique role as voltage sensor for conformational EC coupling is firmly established, the conventional function as $Ca^{2+}$ channel is still enigmatic. Here we show that $Ca^{2+}$ influx via DHPR is not necessary for muscle performance by generating a knock-in mouse where DHPR-mediated $Ca^{2+}$ influx is eliminated. Homozygous knock-in mice display SR $Ca^{2+}$ release, locomotor activity, motor coordination, muscle strength and susceptibility to fatigue comparable to wild-type controls, without any compensatory regulation of multiple key proteins of the EC coupling machinery and $Ca^{2+}$ homeostasis. These findings support the hypothesis that the DHPR-mediated $Ca^{2+}$ influx in mammalian skeletal muscle is an evolutionary remnant.

[1] Division of Biochemical Pharmacology, Department of Medical Genetics, Molecular and Clinical Pharmacology, Medical University of Innsbruck, Peter Mayr Strasse 1, A-6020 Innsbruck, Austria. [2] Institute of Applied Physiology, Ulm University, Albert-Einstein-Allee 11, D-89081 Ulm, Germany. [3] Department of Anaesthesiology, Ulm University, Albert-Einstein-Allee 11, D-89081 Ulm, Germany. Correspondence and requests for materials should be addressed to M.G. (email: manfred.grabner@i-med.ac.at)

Excitation–contraction (EC) coupling in skeletal muscle can be understood only in an evolutionary context. The phylogenetic earliest mechanism involves muscle membrane depolarization, which triggers pore openings of the voltage-gated sarcolemmal L-type $Ca^{2+}$ channel, or dihydropyridine receptor (DHPR), leading to a substantial and fast activating $Ca^{2+}$ inward current[1–3]. This sudden increase in intracellular $Ca^{2+}$ concentration triggers $Ca^{2+}$-sensitive $Ca^{2+}$ release channels or ryanodine receptors (RyR) to open and release massive amounts of $Ca^{2+}$ ions from sarcoplasmic reticulum (SR) stores into the cytoplasm, sufficient to finally induce muscle contraction[4–6]. This phylogenetically ancient EC-coupling mechanism, which is fully dependent on $Ca^{2+}$-induced $Ca^{2+}$ release (CICR), is still extant in skeletal muscles of invertebrates and protochordates[7–12], and interestingly, in all cardiac and smooth muscles up to the evolutionary most derived vertebrates[3, 6, 13].

However, during evolution to vertebrates, novel skeletal-muscle-specific DHPR and RyR isoforms emerged, giving rise to a new and unique mechanism of EC coupling: Vertebrate skeletal muscle EC coupling is based on a $Ca^{2+}$ influx-independent, inter-channel protein–protein interaction between the DHPR and skeletal-muscle-specific RyR type-1 (RyR1)[14]. Upon this physical interaction, the voltage-induced conformational change of the DHPR is transmitted to RyR1, which subsequently opens, releases large amounts of $Ca^{2+}$ from the SR stores and ultimately leads to muscle contraction[15, 16]. This phylogenetically most derived skeletal-muscle-specific mechanism of depolarization-induced $Ca^{2+}$ release combines the advantage of accelerated EC coupling efficacy along with lower energy consumption due to reduced $Ca^{2+}$ extrusion across the sarcolemma[17]. A prerequisite for this $Ca^{2+}$-independent physical DHPR–RyR1 interaction was a transition in the ultrastructural positioning of the DHPRs from random clustering around the RyR (the protochordate/cardiac-type) to an arrangement into tetrads, i.e., four DHPRs positioned strictly adjacent to the homotetrameric RyR1[1, 18].

Interestingly, in addition to their role in conformational EC coupling, mammalian skeletal muscle DHPRs retained their ancient function as $Ca^{2+}$ channels[19]. However, the physiological significance of this DHPR $Ca^{2+}$ current, which is certainly not (immediately) required for EC coupling[14], is still unresolved despite being under investigation since nearly half a century[19]. So far, one can only speculate that this DHPR $Ca^{2+}$ inward current, might contribute in the form of excitation-coupled $Ca^{2+}$ entry (ECCE)[19–22] and have a crucial role in $Ca^{2+}$ homeostasis, e.g., by SR store filling, or is essential for skeletal muscle development via regulation of acetylcholine receptor pre-patterning and formation of neuromuscular junctions[23, 24], or has a negative inotropic effect[25–29], which is crucial for overall muscle health and fibre integrity[29], or finally, is just an evolutionary remnant of the ancient CICR stage[2, 3, 30]. In other words, it is unclear whether the skeletal muscle DHPR $Ca^{2+}$ current is exaptational or vestigial. However, to date it was not possible to answer any of these questions due to the unavailability of an appropriate animal model system with a fully occluded skeletal muscle DHPR.

Recently, a mouse model carrying a mutation in the DHPR selectivity filter (E1014K) was generated[31]. This mutation leads to reduced $Ca^{2+}$ pore-binding affinity and thus changes the $Ca^{2+}$ selectivity of the DHPR by allowing flow of monovalent cations instead of $Ca^{2+}$[32, 33]. The E1014K mouse displays reduced muscle endurance, decreased muscle fibre size and a shift in fibre-type distribution towards fast fibres[31]. The authors concluded that $Ca^{2+}$ permeation and/or binding via/to the DHPR pore plays an important modulatory role in muscle performance[31]. However, a drawback of this interpretation is the negligence of the presence of a persisting permeation of monovalent cations through the mutant EK channel[31, 32, 34], which might, for instance, affect the membrane potential during muscle activity[32].

With the intention to create a mouse model with a fully occluded skeletal muscle DHPR, we used a DHPR pore-blocking mechanism refined by evolution. Previously, we found that zebrafish and all other euteleost fishes express non-$Ca^{2+}$-conducting DHPRs in their skeletal muscles[2, 35]. Consequently, we engineered the selectivity-filter-adjacent point mutation N617D, previously identified to be responsible for DHPR non-conductivity in zebrafish fast, glycolytic muscle[2] into the mouse genome. Electrophysiological investigations on cultured skeletal muscle primary myotubes and adult fibres from the resultant homozygous mutant mouse with non-conducting (nc) DHPR showed complete block of DHPR $Ca^{2+}$ influx but unaltered intracellular SR $Ca^{2+}$ release. Notably, and in contrast to the E1014K model, the ncDHPR pore does not show permeation of monovalent cations[36].

Here we use the ncDHPR mouse model to explore the specific role of the DHPR-mediated $Ca^{2+}$ current in skeletal muscle via a range of muscle performance tests on whole animals and isolated muscles, as well as at the cellular and molecular level with biophysical, immunohistochemical, biochemical and quantitative PCR (qPCR) techniques. Surprisingly, these experiments did not point to any differences between ncDHPR mice and their wild-type (wt) littermates, irrespective of whether mice were young or aged, or the isolated muscles were fast- or slow-twitch type. Overall, our results strongly support the hypothesis that the $Ca^{2+}$ influx through the mammalian skeletal muscle DHPR is irrelevant for muscle performance and consequently can be considered as an evolutionary remnant (vestigial) of the phylogenetic stages of early chordates, where DHPR $Ca^{2+}$ influx was the exclusive trigger for RyR activation in skeletal-muscle EC coupling.

## Results

**Creation of the non-$Ca^{2+}$-conducting (nc)DHPR knock-in mouse.** To thoroughly investigate the physiological relevance of the DHPR $Ca^{2+}$ influx in skeletal muscle, we generated a non-$Ca^{2+}$-conducting (nc) DHPR knock-in mouse. Mutation N617D in pore loop II, previously identified to be responsible for the complete loss of $Ca^{2+}$ conductance in zebrafish DHPR $\alpha_{1S}$-b[2] was introduced into the murine CACNA1S gene (Supplementary Fig. 1a). Strategy and targeting construct used for generating the N617D mutant mouse are shown in Supplementary Fig. 1b. The genotype was confirmed by restriction fragment length polymorphism (RFLP) and sequencing.

**No $Ca^{2+}$ influx but unaltered EC coupling in the ncDHPR mouse.** As a first step, to test whether introduction of the N617D mutation eliminates inward $Ca^{2+}$ currents via the mouse skeletal muscle DHPR, we measured whole-cell $Ca^{2+}$ currents from ncDHPR myotubes isolated from 3 to 4 days old pups and interosseous muscle fibres isolated from young mice. Recordings from ncDHPR myotubes display complete loss of DHPR $Ca^{2+}$ influx (with $I_{max}$ of $-0.02 \pm 0.01$ pA pF$^{-1}$) compared ($P < 0.001$, unpaired $t$-test) with wt myotubes ($I_{max} = -4.87 \pm 0.31$ pA pF$^{-1}$) (Fig. 1a–c). However, voltage dependence and size of intracellular $Ca^{2+}$ release was indistinguishable ($P > 0.05$, unpaired $t$-test) between ncDHPR and wt myotubes (Fig. 1d–f), pointing to intact EC coupling. Next, we examined whether elevating the recording temperature to the physiological value might evoke inward $Ca^{2+}$ currents in ncDHPR myotubes. Interestingly, at ~37 °C, we found a twofold augmentation of $Ca^{2+}$ currents in wt myotubes ($I_{max} = -9.55 \pm 0.70$ pA pF$^{-1}$), but again no inward $Ca^{2+}$ current in ncDHPR myotubes ($I_{max} = -0.07 \pm 0.04$ pA pF$^{-1}$) (Supplementary Fig. 2a). However, contrary to the enhancement of DHPR

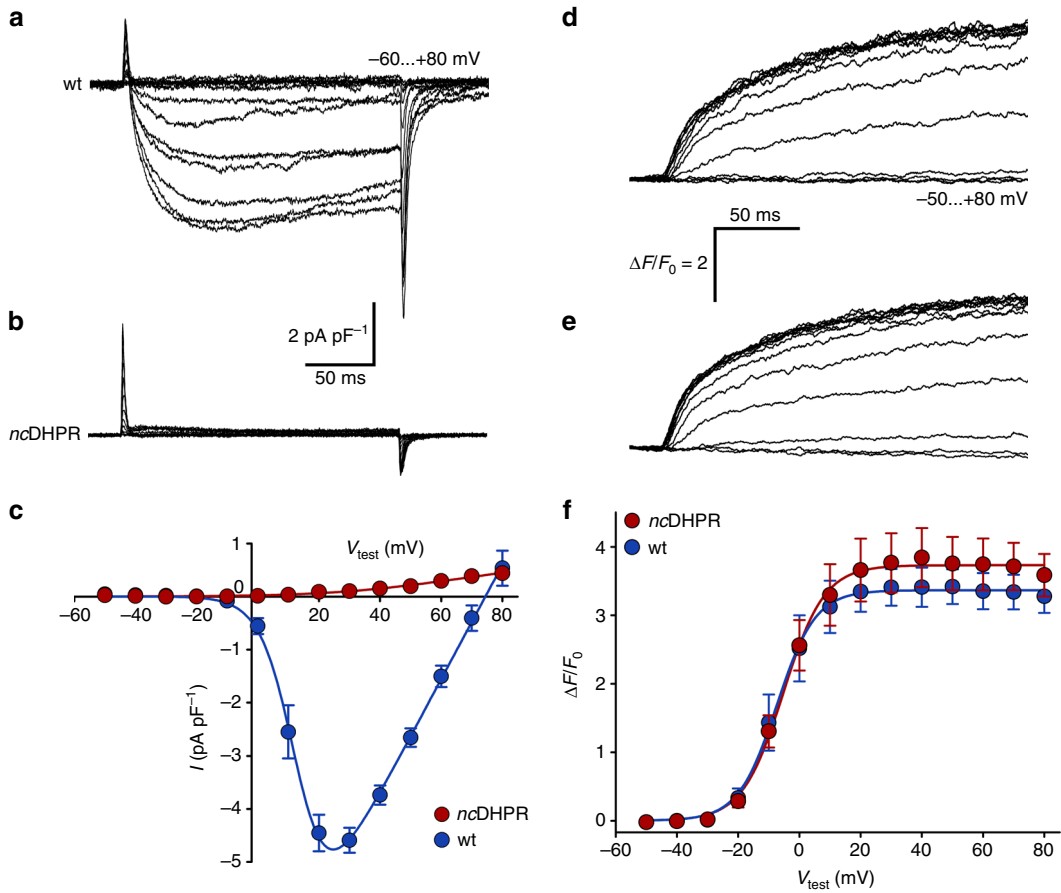

**Fig. 1** N617D mutation completely abolishes DHPR Ca$^{2+}$ influx without altering EC coupling. **a**, **b** Representative whole cell Ca$^{2+}$ current recordings on myotubes isolated from 3 to 4 days old (**a**) wt and (**b**) *nc*DHPR pups, indicating the complete loss of inward Ca$^{2+}$ current in *nc*DHPR mice in contrast ($P < 0.001$) to wt control mice. *Scale bars*, 50 ms (horizontal), 2 pA pF$^{-1}$ (vertical). **c** Current–voltage relationships for DHPR-mediated Ca$^{2+}$ currents recorded from *nc*DHPR ($n = 13$) and wt ($n = 7$) myotubes. **d**, **e** Representative intracellular SR Ca$^{2+}$ release recordings from (**d**) wt and (**e**) *nc*DHPR myotubes. *Scale bars*, 50 ms (horizontal), $\Delta F/F_0 = 2$ (vertical). **f** In contrast to DHPR Ca$^{2+}$ influx, voltage dependence of maximal Ca$^{2+}$ release is indistinguishable ($P > 0.05$) between *nc*DHPR (($\Delta F/F_0$)$_{max} = 3.75 \pm 0.39$; $n = 12$) and wt (($\Delta F/F_0$)$_{max} = 3.39 \pm 0.26$; $n = 8$) myotubes, implying unaltered EC coupling in the *nc*DHPR mouse. Data are represented as mean ± s.e.m.; $P$ determined by unpaired Student's $t$-test

Ca$^{2+}$ influx in wt, there was no effect ($P > 0.05$, unpaired $t$-test) of elevated temperature on intracellular Ca$^{2+}$ release, neither in wt nor *nc*DHPR myotubes (Supplementary Fig. 2b). Consistent with the outcome from primary myotubes, adult interosseous muscle fibres isolated from young *nc*DHPR mice also displayed complete loss of DHPR Ca$^{2+}$ influx ($I_{max} = 0.06 \pm 2.02$ nA) compared ($P < 0.001$, unpaired $t$-test) with wt fibres ($I_{max} = -19.1 \pm 2.62$ nA) (Supplementary Fig. 3). The SR Ca$^{2+}$ release showed the typical phasic-tonic response (Fig. 2a) with indistinguishable ($P > 0.05$, unpaired $t$-test) voltage-dependence of peak Ca$^{2+}$ release flux (Fig. 2b) and Ca$^{2+}$ permeability (Fig. 2c) between *nc*DHPR and wt fibres. As well, kinetic parameters characterizing the shape of the signals viz. time to reach peak activation and the ratio of permeability peak to plateau were comparable between both genotypes at all test potentials (Supplementary Fig. 4). Altogether, these results indicate a complete elimination of skeletal muscle Ca$^{2+}$ influx but unaltered voltage-induced SR Ca$^{2+}$ release in the *nc*DHPR mouse model. Thus, the *nc*DHPR model serves as a unique system for investigating the physiological relevance of the DHPR Ca$^{2+}$ influx in skeletal muscle EC coupling.

**SR Ca$^{2+}$ store content is unaltered in *nc*DHPR mice.** To assess the importance of DHPR Ca$^{2+}$ influx in SR store filling, Fura-FF-AM loaded adult interosseous muscle fibres were challenged with

RyR agonist 4-CmC (4-chloro-*m*-cresol) under conditions that inhibited Ca$^{2+}$ uptake by sarco/endoplasmic reticulum Ca$^{2+}$-ATPase(SERCA) (viz. the addition of CPA, cyclopiazonic acid) (Fig. 3a). SERCA block could be verified by a slow increase in resting Ca$^{2+}$ concentration when CPA was applied alone. Figure 3b depicts the comparison of the peak Fura-FF fluorescence ratios and demonstrates that 4-CmC-induced Ca$^{2+}$ release is not significantly different ($P > 0.05$, unpaired $t$-test) between *nc*DHPR and wt fibres, indicating identical steady state SR Ca$^{2+}$ loading in both genotypes.

**Basic phenotypic characterization of the *nc*DHPR mouse.** The absence of DHPR Ca$^{2+}$ influx in *nc*DHPR mice did not cause embryonic lethality and animals had normal posture and gait, and could not be visually distinguished from wt littermates. No difference ($P > 0.05$, unpaired $t$-test) in male (Fig. 4a) and female (Fig. 4b) body weight development was observed between *nc*DHPR and wt mice. In addition, litter size (Fig. 4b, *inset*) was indistinguishable ($P > 0.05$, unpaired $t$-test) between both genotypes.

To identify possible muscle-type-specific differences we chose extensor digitorum longus (EDL) muscle, which primarily consists of fast-twitch fibres and soleus (SOL) muscle, which is more a slow-twitch muscle compared with EDL. Basic physical parameters such as wet weight, length and diameter of SOL and

EDL muscles, isolated from young (Supplementary Fig. 5a, b) or aged mice (Supplementary Fig. 5c, d) were not different ($P > 0.05$, unpaired $t$-test) between $nc$DHPR and wt mice.

Furthermore, transverse cryosections of SOL and EDL muscle immunostained with antibodies against specific myosin heavy-chain isoforms indicated that the lack of DHPR $Ca^{2+}$ influx in the $nc$DHPR mouse did not induce any change ($P > 0.05$, unpaired $t$-test) in the fibre-type composition, neither in the slower-twitch SOL muscle nor the fast-twitch EDL (Supplementary Fig. 6a, b). Similarly, no significant differences ($P > 0.05$, unpaired $t$-test) in the fibre cross-sectional area (CSA) of SOL and EDL muscles were observed between both genotypes (Supplementary Fig. 6c).

**$nc$DHPR mice show unaltered voluntary locomotor activity**. As the $nc$DHPR mouse did not show any evident basic phenotypic

differences from wt, we next tested for consequences of DHPR $Ca^{2+}$ non-conductivity on voluntary locomotor activity[37] in young (3–6 months old) mice. Home-cage ambulatory activity recorded over 60 h was indistinguishable ($P > 0.05$, unpaired $t$-test) between $nc$DHPR and wt mice (average age: 4.0 months). Both the genotypes displayed normal behaviour, with higher activity during the dark cycle and lower activity during the light cycle (Fig. 5a). Furthermore, to record enhanced voluntary locomotor activity[37], the mice (average age: 4.1 months) were challenged with a voluntary activity wheel experiment for 21 days. Consistent with above results, the cumulative average distance covered by $nc$DHPR mice ($19.29 \pm 6.31$ km) was similar ($P > 0.05$, unpaired $t$-test) to that of wt control mice ($21.28 \pm 3.71$ km) (Fig. 5b).

**Ex vivo contractile properties are unaltered in $nc$DHPR mice**. To test the impact of the loss of DHPR $Ca^{2+}$ influx on skeletal muscle functioning in intact muscles independent of other parameters and under high frequency stimulations[38], we compared ex vivo contractile properties, viz. force and fatigability of SOL and EDL muscles isolated from $nc$DHPR and wt mice. For this part of the study, young (3–7 months old; average age: 4.4 months) and aged (17–22 months old; average age: 19.3 months) mice were used to examine putative age-related accumulative degenerative effects on muscle performance due to the loss of DHPR $Ca^{2+}$ influx. On being subjected to force-frequency tests, SOL and EDL muscles did not show any

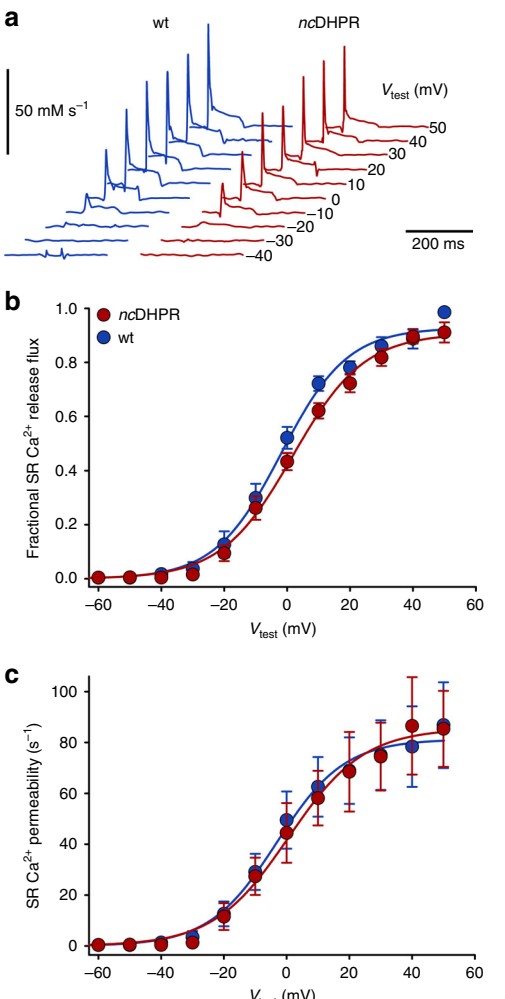

**Fig. 2** Unaltered voltage dependence of SR $Ca^{2+}$ release in $nc$DHPR adult muscle fibres. **a** Representative traces of $Ca^{2+}$ release flux signals determined from isolated adult toe fibres (*musculus interosseus*) at 100-ms depolarizing voltage steps. The flux exhibits an early peak and a rapid decline to a lower level (plateau). *Scale bars*, 200 ms (horizontal), 50 mM $s^{-1}$ (vertical). **b** Voltage dependence of normalized peak $Ca^{2+}$ release flux is indistinguishable ($P > 0.05$) between $nc$DHPR ($n = 12$) and wt ($n = 10$) adult muscle fibres. **c** $Ca^{2+}$ release flux traces were converted to permeability with the assumption that the slow decline during the plateau phase results from SR depletion[60]. The calculated peak $Ca^{2+}$ release permeability as a function of voltage showed no significant difference ($P > 0.05$) between $nc$DHPR and wt fibres. Data are represented as mean $\pm$ s.e.m.; $P$ determined by unpaired Student's $t$-test

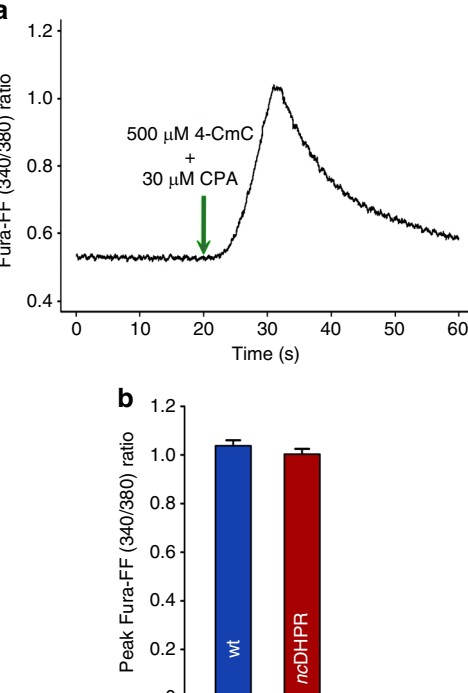

**Fig. 3** Absence of DHPR-mediated $Ca^{2+}$ influx has no impact on SR $Ca^{2+}$ store content. **a** Representative trace of SR $Ca^{2+}$ release in intact interosseous muscle fibres triggered by application (green arrow) of 500 µM of RyR agonist 4-CmC in the presence of 30 µM of SERCA blocker CPA. The resulting $Ca^{2+}$ signals were measured with the low affinity indicator Fura-FF-AM to avoid dye saturation during $Ca^{2+}$ release stimulation. **b** Comparison of 4-CmC-induced $Ca^{2+}$ signals, measured as peak fluorescence ratio of Fura-FF (340/380) showed no significant difference ($P > 0.05$) between $nc$DHPR ($1.0 \pm 0.02$; $n = 53$) and wt muscle fibres ($1.04 \pm 0.02$; $n = 48$), indicating identical SR $Ca^{2+}$ store filling. Bars represent mean $\pm$ s.e.m.; $P$ determined by unpaired Student's $t$-test

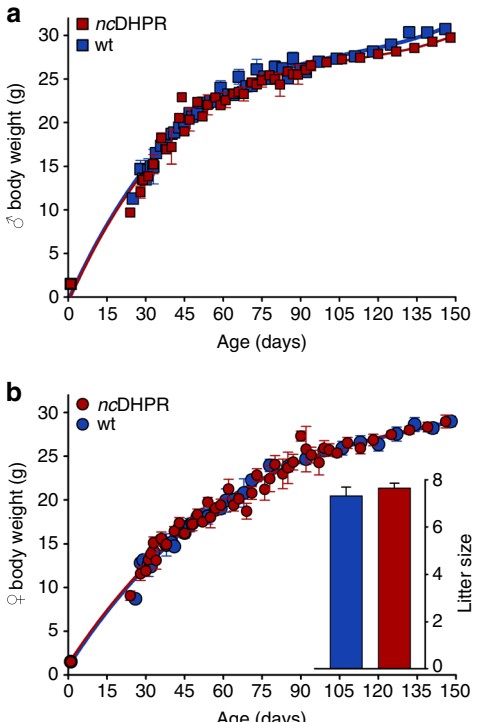

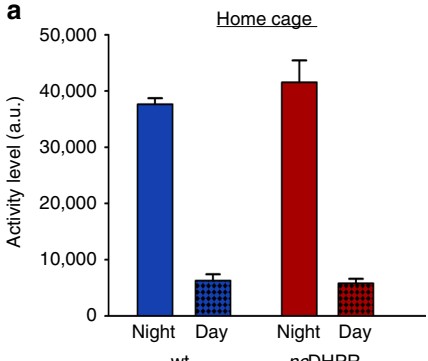

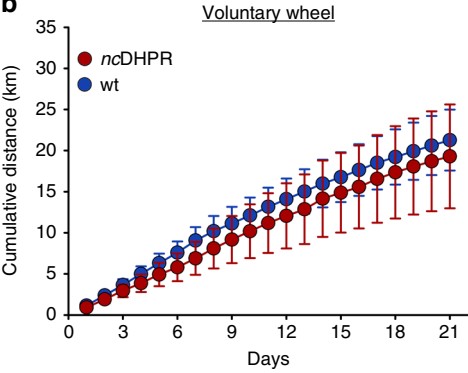

**Fig. 4** Lack of DHPR Ca²⁺ influx has no impact on body weight development and fertility. Growth curve of (**a**) male and (**b**) female *nc*DHPR and wt mice was monitored weekly for 150 days. **a** Male *nc*DHPR ($n = 30$) and wt ($n = 35$) siblings showed identical ($P > 0.05$) body weight development. **b** Similarly, female *nc*DHPR ($n = 29$) showed no difference ($P > 0.05$) in body weight development compared with their wt control siblings ($n = 31$). At the end of the observation period, the maximum body weight of male *nc*DHPR and wt mice was $29.76 \pm 0.55$ and $30.74 \pm 0.54$ g, respectively, and of female *nc*DHPR and wt mice was $28.96 \pm 0.73$ and $28.98 \pm 0.60$ g, respectively. **b** (*Inset*) The mean litter size of *nc*DHPR mice ($7.64 \pm 0.21$; $n = 100$) was similar ($P > 0.05$) to wt mice ($7.31 \pm 0.37$; $n = 49$). Data are represented as mean ± s.e.m.; $P$ determined by unpaired Student's *t*-test

**Fig. 5** Voluntary locomotor activity is identical in *nc*DHPR and wt mice. **a** Home cage ambulatory activity was measured for 2 days and 3 nights with an infrared video monitoring system. Young (3–6 months old) *nc*DHPR mice ($n = 10$) displayed high nighttime activity ($41,520 \pm 3,934$ a.u.) and low daytime activity ($5,794 \pm 790$ a.u) similar ($P > 0.05$) to the control wt mice ($37,620 \pm 1,079$ a.u. and $6,247 \pm 1,152$ a.u., respectively; $n = 6$). **b** Average cumulative distance covered during 21 days of voluntary activity wheel task (binned every 24 h) was indistinguishable ($P > 0.05$) between young *nc*DHPR ($n = 18$) and wt ($n = 17$) mice. Data are represented as mean ± s.e.m.; $P$ determined by unpaired Student's *t*-test

differences ($P > 0.05$, unpaired *t*-test) in maximum twitch force between young *nc*DHPR and wt mice (Fig. 6a, *left graph*). Furthermore, the maximum twitch force observed in both muscle types of aged mice was indistinguishable ($P > 0.05$, unpaired *t*-test) between *nc*DHPR and wt (Fig. 6a, *right graph*). The tetanic force exhibited by SOL and EDL at increasing stimulation frequencies (for exemplar recordings see Supplementary Fig. 7a, b) were similar ($P > 0.05$, unpaired *t*-test) in young (Fig. 6b, *left graph*) or aged (Fig. 6b, *right graph*) *nc*DHPR and wt mice.

As the EC coupling mechanism is affected in muscle fatigue[38], the implications of the loss of DHPR Ca²⁺ influx in the *nc*DHPR mouse might become evident only under extreme fatiguing conditions. Hence, we subjected isolated SOL and EDL muscles to fatigue-inducing repetitive high-frequency tetanic stimulations (Supplementary Fig. 7c, d). SOL is markedly more fatigue resistant than EDL due to its higher oxidative capacity[38]. Consistent with the above findings, susceptibility to fatigue was not different ($P > 0.05$, unpaired *t*-test) between muscles isolated from young *nc*DHPR and wt mice (Fig. 6c, *left graph*). Likewise, there was no significant difference in fatigability between muscle derived from aged *nc*DHPR and wt mice (Fig. 6c, *right graph*). Akin to the physiological muscle parameters, kinetic parameters of single twitches such as time to peak, time to half-relaxation and twitch duration were similar ($P > 0.05$, unpaired *t*-test) in muscles from *nc*DHPR and wt, regardless of whether isolated from young or aged mice (Supplementary Fig. 8).

**_nc_DHPR mice display normal muscle coordination and strength**. DHPR-mediated Ca²⁺ influx has no relevance in muscle force production, as can be concluded from the above ex vivo isometric contraction experiments on isolated SOL and EDL muscles from *nc*DHPR and wt mice. However, to assess whether this also holds true at the whole animal level, we confronted the animals with a battery of tests to measure muscle coordination and strength[37]. In the beam walk test, the average time to traverse the 120-cm-long beam was similar ($P > 0.05$, unpaired *t*-test) in young *nc*DHPR ($8.40 \pm 0.31$ s; $n = 10$) and wt ($8.83 \pm 0.60$ s; $n = 6$) mice. Next, we compared muscle strength of all four limbs[37] by letting the mice hang upside down on a cross-wired grid for a maximum of 600 s. The latency to fall from the inverted grid was indistinguishable ($P > 0.05$, unpaired *t*-test) between young *nc*DHPR and wt control mice (Fig. 7a, *left two bars*). Mice that achieved the maximum hanging time were also similar for both genotypes (14 *nc*DHPR and 12 wt mice). Likewise, no significant difference ($P > 0.05$, unpaired *t*-test) in performance was observed between aged *nc*DHPR and wt mice (Fig. 7a, *right two bars*). None of the aged mice reached the maximum hanging time of 600 s. Finally, we performed a highly specific grip strength test to exclude other aspects of behaviour and coordination[37], and exclusively measure the maximal muscle strength of forelimbs. In this test, the mouse was allowed to grasp onto a trapezoidal steel grip and after a gentle pull, the maximal force (in g) exerted before it releases the grip was recorded. Consistent with results

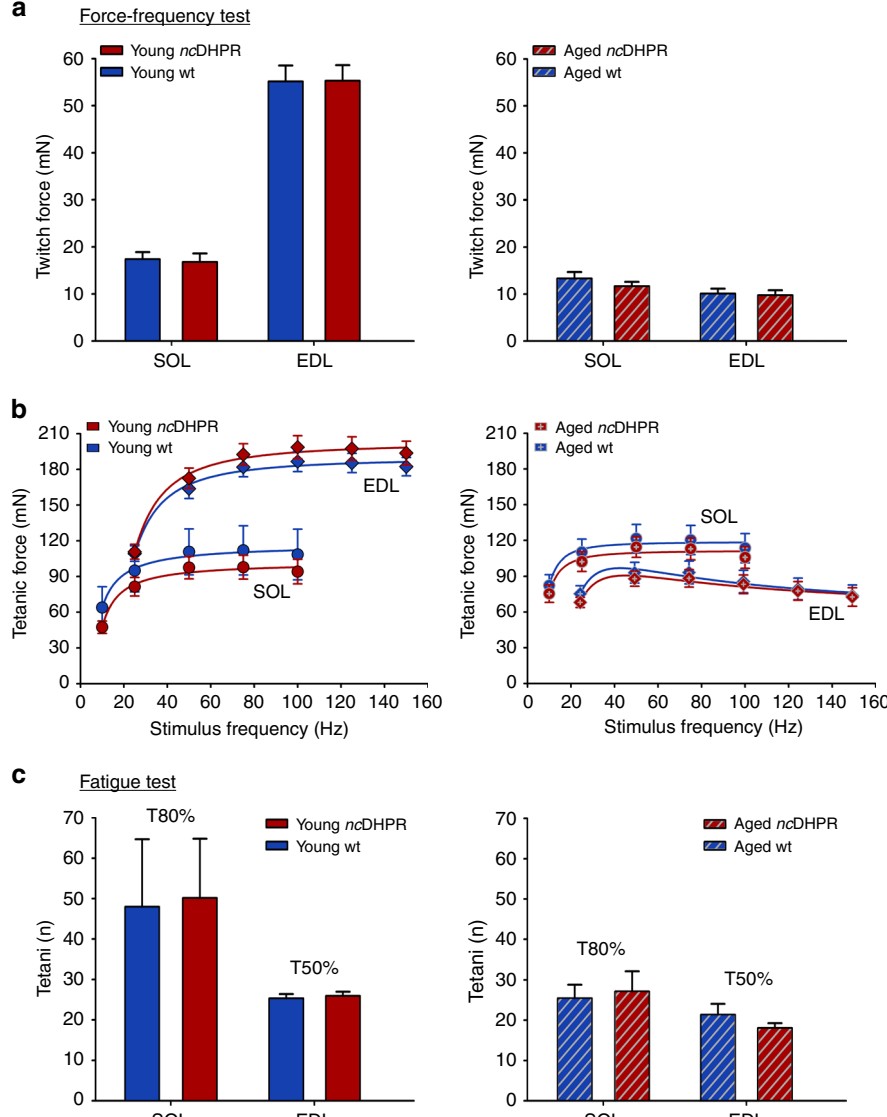

**Fig. 6** Absence of DHPR Ca$^{2+}$ influx has no impact on ex vivo muscle force and fatigue. Soleus (SOL) and extensor digitorum longus (EDL) muscles isolated from young and aged *nc*DHPR and wt control mice were subjected to (**a**, **b**) force-frequency and **c** repetitive tetanic fatigue tests (see also Supplementary Fig. 7). **a** The average maximum twitch force (at supramaximal voltage, 25 V for 1 ms) is identical ($P > 0.05$) in SOL and EDL muscles between young (*left graph*) *nc*DHPR (SOL: 16.78 ± 1.80 mN; $n = 12$; EDL: 55.33 ± 3.31 mN; $n = 19$) and wt (SOL: 17.39 ± 1.49 mN: $n = 9$; EDL: 55.14 ± 3.39 mN: $n = 18$) mice, as well as between aged (*right graph*) *nc*DHPR (SOL: 11.66 ± 0.91 mN; $n = 19$ and EDL: 9.76 ± 1.04 mN; $n = 20$) and wt (SOL: 13.33 ± 1.35 mN; $n = 18$ and EDL: 10.09 ± 1.06 mN; $n = 22$) mice. **b** Frequency dependence of average tetanic force in SOL and EDL muscles is also unaltered ($P > 0.05$) between age-matched *nc*DHPR and wt mice (*left* and *right graphs*). **c** Under high-frequency repetitive stimulations, the number of tetani till the tetanus amplitude decreased to 80% (T80%) of the initial force in case of SOL and 50% (T50%) for EDL was taken as an index of fatigue. No difference in susceptibility to fatigue was observed in SOL and EDL muscles isolated from either young (*left graph*) *nc*DHPR (SOL: T80% = 50.12 ± 14.65 n; $n = 12$; EDL: T50% = 25.95 ± 1.04 n; $n = 19$) and wt (SOL: T80% = 48.0 ± 16.66 n; $n = 9$; EDL: T50% = 25.33 ± 1.05 n; $n = 18$) mice or aged (*right graph*) *nc*DHPR (SOL: T80% = 27.22 ± 4.89 n; $n = 18$; EDL: T50% = 18.12 ± 1.14 n; $n = 17$) and wt (SOL: T80% = 25.5 ± 3.31 n; $n = 18$; EDL: T50% = 21.42 ± 2.61 n; $n = 19$) mice. All recordings were performed at room temperature (~26 °C). Data are represented as mean ± s.e.m.; $P$ determined by unpaired Student's $t$-test

obtained with other tests, young and aged *nc*DHPR mice displayed similar ($P > 0.05$, unpaired $t$-test) muscle strength to their age-matched control wt siblings (Fig. 7b). Average body weight was homogenous across all the cohorts.

**Similar whole animal muscle endurance/fatigue in *nc*DHPR mice.** Ex vivo measurements on isolated SOL and EDL muscles showed no alterations in fatigability in the absence of DHPR-mediated Ca$^{2+}$ influx in *nc*DHPR mice. To confirm these findings and fully characterize the skeletal muscle functioning in its

physiological environment in the *nc*DHPR mice, we performed involuntary fatigue and endurance tests at the whole animal level. As a first measure of sensorimotor skills and fatigue, animals were confronted to a standard Rotarod test with steady acceleration[37] from 4 to 40 rpm over 300 s. Young and aged *nc*DHPR mice remained on the accelerating rod for the same ($P > 0.05$, unpaired $t$-test) time such as the age-matched wt control mice (Fig. 8a). Likewise, the maximum acceleration at which the mice fell from the rotating rod was indistinguishable ($P > 0.05$, unpaired $t$-test) between young *nc*DHPR and wt mice (26.74 ± 1.67 rpm and 25.03 ± 1.79 rpm, respectively). Consistently, aged *nc*DHPR and

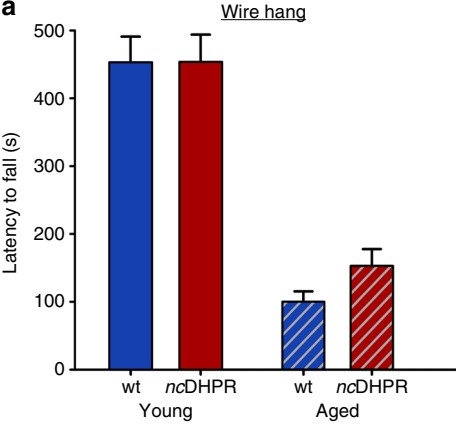

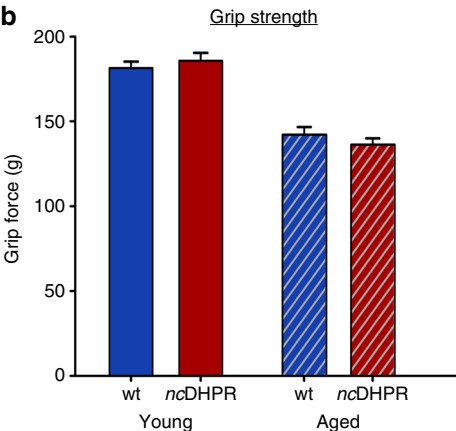

**Fig. 7** DHPR Ca$^{2+}$ influx has no impact on whole animal muscle coordination and strength. **a** ncDHPR mice performed equally well similar to control littermates in wire hang test set for 600 s. The mean latency to fall from the cross-wired grid was similar between young ncDHR (453.85 ± 39.97 s; $n = 26$) and wt (453.04 ± 38.06 s; $n = 25$) mice, as well as between aged ncDHPR (153.12 ± 24.82 s; $n = 25$) and wt (100.44 ± 15.06 s; $n = 27$) mice. **b** The mean grip strength, an index of maximal muscle force of forelimbs, was not different in young (185.68 ± 4.71 g; $n = 15$) and aged (136.39 ± 3.59 g; $n = 23$) ncDHPR mice compared with their age-matched wt littermates (young wt: 181.38 ± 3.88 g; $n = 6$; aged wt: 142.29 ± 4.34 g; $n = 27$ mice). Normal age-dependent deterioration in muscle strength is evident in both genotypes. Bars represent mean ± s.e.m.; $P$ determined by unpaired Student's $t$-test

wt mice displayed equal performance (20.07 ± 0.81 rpm and 20.01 ± 1.22 rpm, respectively). Next, we designed a prolonged Rotarod protocol (600 s) at constant speed (20 rpm for young and 15 rpm for aged mice) to access endurance. Latency to fall displayed by young ncDHPR mice was comparable ($P > 0.05$, unpaired $t$-test) to their wt counter mates (Fig. 8b, *right two bars*). Likewise, endurance level was indistinguishable ($P > 0.05$, unpaired $t$-test) between aged mice of both genotypes (Fig. 8b, *left two bars*).

Resistance to fatigue is an optimal, comprehensive measure of muscle performance and one of the most powerful tools for analysing fatigue resistance is a treadmill task. An acute exhaustive treadmill protocol was used to measure the enforced maximum endurance limit of ncDHPR mice[39]. Interestingly, young mice of both genotypes accomplished the entire 0.9 km run with similar efficiency. The total average resting time cumulated during the 1-h task, as well as the cumulative number of rests at the end of the task were indistinguishable ($P > 0.05$, unpaired $t$-test) between both genotypes (Fig. 8c). Finally, due to age-

related depreciation in muscle strength (see Fig. 6), aged mice were challenged with a milder protocol (see Methods). Consistent with the above findings, aged ncDHPR mice completed the entire 0.55 km treadmill run in 1 h with similar ($P > 0.05$, unpaired $t$-test) competence to their age-matched wt siblings (Fig. 8d). The comparable total average cumulative resting time reached by ncDHPR and wt mice indicates no age-related accumulative adverse effects on skeletal muscle fatigability despite the life-long absence of DHPR Ca$^{2+}$ influx in ncDHPR mice. Akin to young mice, there was again no difference ($P > 0.05$, unpaired $t$-test) in the cumulative number of rests between aged ncDHPR and wt mice. Altogether, results from all the above experiments indicate that ncDHPR mice exhibit equivalent muscle performance to their wt siblings, which convincingly points to the insignificance of DHPR Ca$^{2+}$ influx for skeletal muscle function.

**No compensatory regulation of triad proteins in ncDHPR mice.** The absence of DHPR Ca$^{2+}$ influx has no obvious impact on general physiological parameters and whole animal muscle performance, neither in young nor aged ncDHPR mice. Apparently, the question arose whether up- or downregulation of any of the key triadic proteins responsible for either influx or efflux of Ca$^{2+}$ ions compensates for the lack of DHPR-mediated Ca$^{2+}$ influx in the ncDHPR mice. To address this issue, we first opted for qPCR analysis to quantify the messenger RNA levels transcribed from a selected panel of genes encoding for proteins involved in EC coupling and Ca$^{2+}$ homeostasis, with β-actin (ACTB) and eukaryotic translation elongation factor 1 alpha 2 (EEF1A2) as reference genes. Total RNA was isolated from skeletal muscle of mice at different developmental stages viz. neonatal mice and adult SOL and EDL muscles. Starting with the main players of EC coupling, DHPR and RyR1, we found no differences ($P > 0.05$, unpaired $t$-test) in transcript levels between neonatal ncDHPR and wt mice (Fig. 9a). This lack of compensatory transcriptional DHPR upregulation is congruent with the density of DHPR Ca$^{2+}$ currents recorded from heterozygous wt/ncDHPR myotubes, which did not reach more than half the current size compared ($P < 0.001$, unpaired $t$-test) with that recorded from wt myotubes ($I_{max} = -2.39 \pm 0.15$ and $-4.87 \pm 0.31$ pA pF$^{-1}$, respectively; Supplementary Fig. 9). Furthermore, expression profiles of mRNAs that code for critical Ca$^{2+}$-handling proteins viz. TRPC1, Orai (predominant isoforms Orai1, Orai2 and Orai3), STIM1, SERCA1, PMCA1, NCX (isoforms NCX1 and NCX3) and CSQ (isoforms CSQ1 and CSQ2) were unaltered in skeletal muscles of neonatal ncDHPR mice compared ($P > 0.05$, unpaired $t$-test) with wt mice (Fig. 9). All PCRs were performed in triplicates on 18 first-strand replicates from 9 pups, with EEF1A2 as reference gene. Comparable results were obtained with ACTB as the reference gene (Supplementary Table 1). Likewise, there was no significant difference ($P > 0.05$, unpaired $t$-test) at the transcriptional level of all the above mentioned proteins in adult SOL or EDL muscles between both genotypes with EEF1A2 as reference gene (Supplementary Fig. 10).

Subsequently, to assess putative changes at the translational level, we implemented quantitative western blot analysis of selected key triadic EC-coupling proteins in tibialis anterior (TA) muscles derived from adult ncDHPR and wt mice. Densitometric analyses of immunoreactive bands with glyceraldehyde 3-phosphate dehydrogenase (GAPDH) as internal reference, indicated comparable protein levels ($P > 0.05$, unpaired $t$-test) between both genotypes (Fig. 9b and Supplementary Figs. 11 and 12). Thus, our results clearly indicate that no adaptive transcriptional or translational regulation of key Ca$^{2+}$-handling proteins occurs in the ncDHPR mouse, which could account for its unaltered muscle performance.

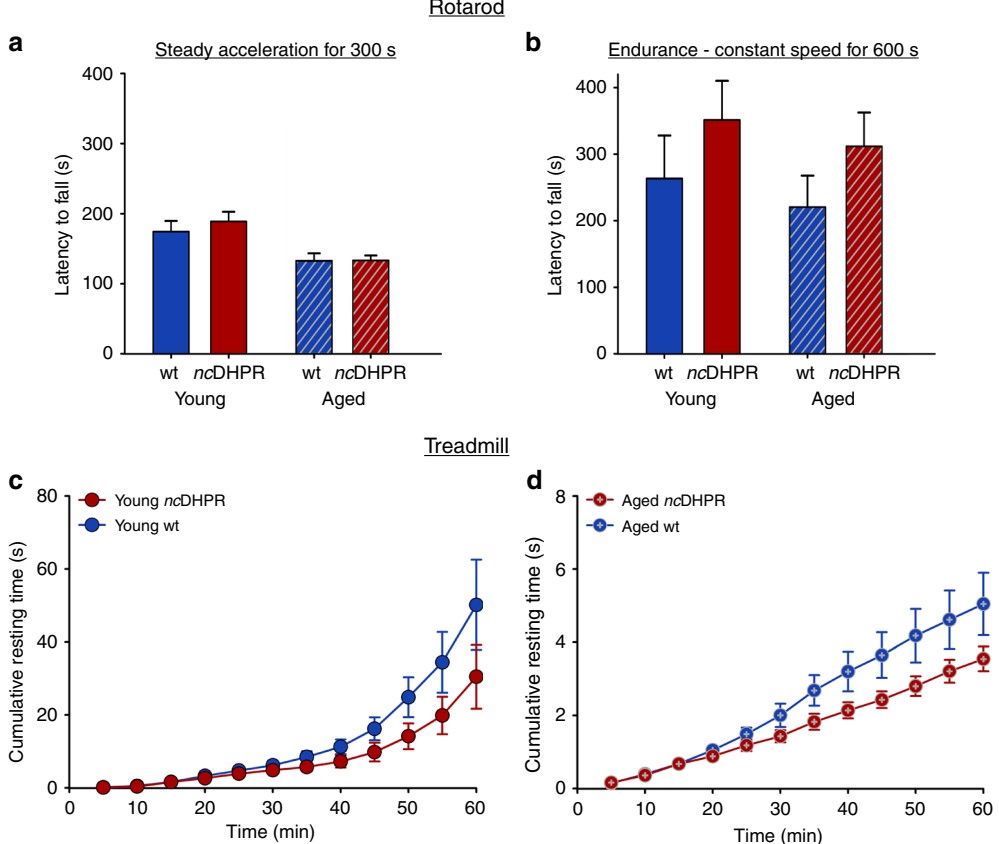

**Fig. 8** DHPR $Ca^{2+}$ influx has no impact on whole animal muscle endurance and fatigue. **a** In Rotarod performance test, the mean latency to fall from a rotating rod under steady acceleration (4–40 rpm for 300 s) was indistinguishable ($P > 0.05$) when young ($188.78 \pm 13.95$ s; $n = 16$) and aged $nc$DHPR ($133.36 \pm 6.73$ s; $n = 26$) mice were compared with their wt ($174.50 \pm 15.02$ s; $n = 16$ and $132.85 \pm 10.19$ s; $n = 28$, respectively) counter mates. **b** The mean latency to fall during 600-s Rotarod endurance task at constant speed, 20 rpm or 15 rpm in young or aged mice, respectively, was indifferent ($P > 0.05$) between age-matched $nc$DHPR (young: $351.54 \pm 58.64$ s; $n = 16$ and aged: $312.22 \pm 50.76$ s; $n = 25$) and wt (young: $263.43 \pm 64.66$ s; $n = 16$ and aged: $220.67 \pm 47.32$ s; $n = 27$) mice. The number of mice that remained on the rotating rod for the entire duration of the task was similar for both genotypes (5 mice each for the young group and 10 aged $nc$DHPR versus 6 aged wt mice). **c** Mean cumulative resting time during an intense treadmill run (0.9 km in 1 h) in young $nc$DHPR ($30.45 \pm 8.75$ s; $n = 17$) was similar ($P > 0.05$) to wt control ($50.15 \pm 12.36$ s; $n = 17$) mice, indicating unaltered susceptibility to muscle fatigue and endurance. The cumulative number of rests at the end of the task was also indistinguishable ($P > 0.05$) between both genotypes ($139.94 \pm 37.97$ for $nc$DHPR and $232.18 \pm 52.89$ for wt mice). **d** Mean cumulative resting time during 1-h treadmill run (0.55 km) in aged $nc$DHPR mice ($3.54 \pm 0.34$ s; $n = 23$) was analogous to the wt ($5.04 \pm 0.85$ s; $n = 27$) siblings. Cumulative number of rests at the end of the task was again comparable ($P > 0.05$) between aged $nc$DHPR and wt mice ($19.65 \pm 2.27$ s and $24.52 \pm 3.82$ s, respectively). Data are represented as mean $\pm$ s.e.m.; $P$ determined by unpaired Student's $t$-test

Overall, the surprising outcome of our study from the plethora of in vitro, ex vivo and whole-animal tests underlines the fact that DHPR $Ca^{2+}$ influx is irrelevant for proper skeletal muscle functioning.

**DHPR $Ca^{2+}$ influx does not reduce skeletal muscle contraction.** Previous work proposed[29] that DHPR $Ca^{2+}$ influx has a negative regulatory effect on skeletal muscle contraction by inhibition of the $Ca^{2+}$-sensitive adenylyl cyclase (AC) isoforms, AC5 and AC6. Apparently, our knock-in $nc$DHPR mouse provides the perfect system to re-investigate this hypothesis. As intracellular SR $Ca^{2+}$ release was unaltered in $nc$DHPR mice (refer Fig. 1d–f) in contrast to the described augmentation of $Ca^{2+}$ release[29], we first compared expression levels of AC5 and AC6 RNAs by qPCR assay with *EEF1A2* as reference gene. Conversely, we did not find any adaptive transcriptional downregulation of either of the AC isoforms in skeletal muscles of neonatal $nc$DHPR mice compared ($P > 0.05$, unpaired $t$-test) with wt mice (Supplementary Fig. 13a). Likewise, there was no significant difference ($P > 0.05$, unpaired

$t$-test) in expression levels of AC5 and AC6 in adult SOL and EDL muscles between both genotypes (Supplementary Fig. 10). The negative inotropic effect of $Ca^{2+}$ influx was demonstrated in the diaphragm muscle by ex vivo blocking of DHPRs with the chemically distinct L-type $Ca^{2+}$ channel blockers, nifedipine and verapamil[29]. Using an analogous approach, we compared the isometric contractility of diaphragm muscles, isolated from $nc$DHPR and wt mice, after treatment with the above mentioned drugs. Consistent with the previous findings[29], 10 µM of nifedipine or 50 µM of verapamil induced a robust increment in muscle twitch contraction force in wt mice (Supplementary Fig. 13b, c, respectively). Surprisingly, and in stark contrast to the previous hypothesis[29], application of 10 µM nifedipine to the $nc$DHPR diaphragm muscle also exhibited ~ 30% increase in isometric contraction force, which was indistinguishable ($P > 0.05$, unpaired $t$-test) from control wt muscle (Supplementary Fig. 13b). Similarly, the positive inotropic effect of verapamil (~ 45% increment in twitch force) was indistinguishable ($P > 0.05$, unpaired $t$-test) between $nc$DHPR and wt diaphragm muscles (Supplementary Fig. 13c). The increment in twitch force

with both drugs could be reversed by washing. Altogether, in stark contrast to the previous findings[29], our data from the *nc*DHPR mouse model provide compelling evidence that DHPR $Ca^{2+}$ influx does not attenuate skeletal muscle contraction.

## Discussion

In this study we investigated the physiological consequences of the elimination of skeletal muscle inward $Ca^{2+}$ current in our newly generated *k.i.*-mutant mouse model (*nc*DHPR), which expresses a non-$Ca^{2+}$-conducting (*nc*) DHPR in skeletal muscle. Despite more than 40 years of research, the role of this $Ca^{2+}$

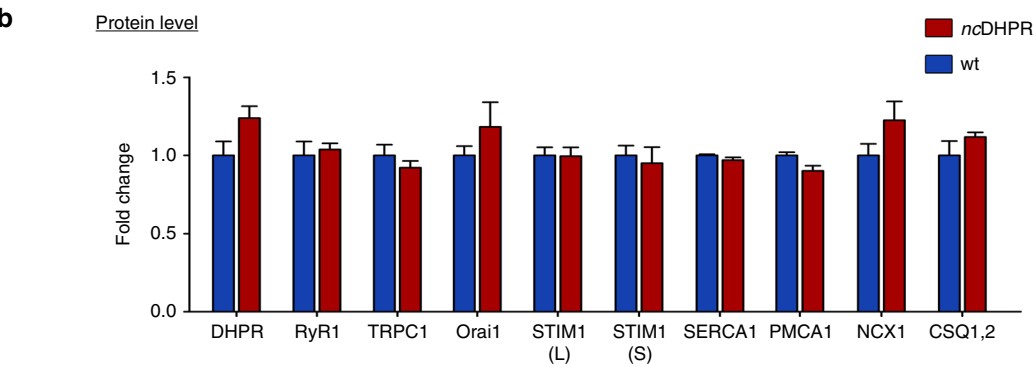

influx in mammalian skeletal muscle which occurs during conformational EC coupling is still obscure and controversial[19]. Theoretically, there are three possible roles for this enigmatic DHPR Ca$^{2+}$ influx: (i) it still marginally contributes to skeletal muscle EC coupling via a rudimentary CICR effect; (ii) it is important as an exaptational current, i.e., during evolution it underwent a change in function from the crucial initiator of CICR to a Ca$^{2+}$ current that serves another function now; or lastly, (iii) it is just a vestigial current, viz., an evolutionary remnant Ca$^{2+}$ current from phylogenetically ancestral CICR stages[3, 7–12] but without any functional importance.

Nevertheless, contemplating these three options it seems very unlikely that this DHPR Ca$^{2+}$ current still contributes to EC coupling via rudimentary CICR, because it is small, slowly activating and considerably lags behind the activation of SR Ca$^{2+}$ release[40, 41]. Moreover, previous biophysical experiments on skeletal muscles ex vivo, muscle fibres and myotubes from frogs to rodents, explicitly showed that pharmacological blocking of the DHPR or elimination of extracellular Ca$^{2+}$ did not obstruct EC coupling[14, 42–45] and, consequently, it is widely accepted that the DHPR Ca$^{2+}$ influx is not an (immediate) prerequisite for the contractile activation[19, 30].

However, substantial arguments against a complete and long-term insignificance of the DHPR Ca$^{2+}$ influx (i.e., being vestigial) also exist, because it is apparently regulated in different ways. For example, the accessory DHPRα$_2$δ-1 subunit functions as major determinant of the characteristic slow L-type Ca$^{2+}$ current kinetics[46, 47] and the DHPRγ$_1$ acts similar to an endogenous Ca$^{2+}$ antagonist that reduces Ca$^{2+}$ entry under conditions promoting modest membrane depolarization[48]. DHPR Ca$^{2+}$ influx is also massively (~8-fold) enhanced in developing fetal mouse skeletal muscle due to the expression of an alternatively spliced DHPRα$_{1S}$ subunit[49, 50] and even more substantially amplified (~30-fold) by retrograde RyR1-DHPR coupling[51, 52]. Hence, the probability of this regulated skeletal muscle DHPR Ca$^{2+}$ influx to function as an exaptational Ca$^{2+}$ current cannot be ruled out. In this case its character would have completely changed, from a crucial EC coupling signal transmitter to a supporter of muscle cell Ca$^{2+}$ homeostasis (e.g., SR store filling), tightly linked to the EC coupling process.

Eventually, to gain deeper and clearer insight into the role of DHPR Ca$^{2+}$ conductivity in mammalian skeletal muscle, we engineered point mutation N617D from pore loop II of zebrafish DHPRα$_{1S}$-b, identified to be responsible for the complete lack of Ca$^{2+}$ conductivity in fast, glycolytic muscle[2], into the murine gene CACNA1S (Supplementary Fig. 1). Whole-cell Ca$^{2+}$ current measurements revealed that indeed this novel ncDHPR mouse model expresses a non-conducting Ca$^{2+}$ channel in skeletal muscle (Fig. 1 and Supplementary Figs. 2 and 3). Survival of the ncDHPR mice and thriving into adulthood provided us a unique

opportunity to study muscle physiology, development and performance in the absence of DHPR Ca$^{2+}$ influx in young and aged skeletal muscle. Elimination of skeletal muscle DHPR Ca$^{2+}$ influx might cause an imbalance in the intracellular Ca$^{2+}$ levels. This would consequently lead to reduced muscle performance, which on the long term (with aging) might induce gross physiological effects such as myotonia or paralysis. However, very surprisingly, our results clearly demonstrate that the absence of the physiological DHPR Ca$^{2+}$ influx in ncDHPR mice is irrelevant for SR store filling (Fig. 3). DHPR Ca$^{2+}$ influx, in particular the massive conductance via the embryonic splice variant of the DHPRα$_{1S}$ subunit, has also been proposed to regulate acetylcholine receptor pre-patterning and formation of neuromuscular junction and thus to be crucial for skeletal muscle development[23, 24]. Contrary to this proposal, we found no motor coordination defects in ncDHPR mice. Notably, our findings also provide clear and firm evidences that the lack of DHPR Ca$^{2+}$ influx in ncDHPR mice is inconsequential for body weight development, fertility, locomotor activity, muscle fibre-type composition, fibre CSA, muscle strength and susceptibility to fatigue, irrespective of the age. Thus, we can undoubtedly exclude putative age-related accumulative effects of the lack of DHPR Ca$^{2+}$ influx on overall muscle health and performance. In addition, a reduction to half the size of whole-cell Ca$^{2+}$ current density in heterozygous wt/ncDHPR mice compared with wt (Supplementary Fig. 9), points to a lack of compensatory upregulation of the DHPR activity. Furthermore, qPCR and quantitative western blot analyses of multiple key proteins of the EC coupling machinery and Ca$^{2+}$ homeostasis did not indicate any transcriptional or translational up- or down-regulation to compensate for the lack of DHPR Ca$^{2+}$ influx in ncDHPR mice (Fig. 9 and Supplementary Figs. 10–12). This is especially interesting for the proteins involved in SR Ca$^{2+}$ store refilling via store-operated Ca$^{2+}$ entry (SOCE) such as TRPC, STIM and Orai[53], because it shows that despite the lack of ECCE in the ncDHPR mouse muscle the SOCE machinery is not upregulated to compensate for the loss of ECCE. However, analyses of mRNA or protein levels of these major players in EC coupling and Ca$^{2+}$ homeostasis do not necessarily equate to their activity levels since a number of these proteins are post-translationally regulated. Still, the evidence that these key proteins remain uninfluenced by the lack of DHPR Ca$^{2+}$ influx is very high.

ECCE was described as store-independent Ca$^{2+}$ entry pathway[20] via the skeletal muscle DHPR[22], activated by repetitive or prolonged depolarizations and to be essential for store repletion[20, 53]. The complete lack of Ca$^{2+}$ influx in ncDHPR mouse myotubes even after 2-s prolonged depolarizations in the presence of the 1,4-dihydropyridine (DHP) agonist BayK8644[36], presents convincing evidence that skeletal muscle of the ncDHPR mouse does not require ECCE for its proper functioning.

**Fig. 9** ncDHPR mice do not show compensatory regulation of key triadic EC coupling proteins. **a** Schematic representation of the skeletal muscle triad with T-tubular invagination (T-tubule) of the sarcolemma adjacent to the sarcoplasmic reticulum Ca$^{2+}$ store (SR). TaqMan qPCR assay (comparative $C_T$ method) on skeletal muscle from neonatal mice revealed that none of the investigated key EC coupling and Ca$^{2+}$ handling proteins is transcriptionally up- or down-regulated ($P > 0.05$) in ncDHPR mice ($n = 18$) compared to wt controls ($n = 18$). All PCRs were performed in triplicates on 18 first strand replicates from 9 pups. For each gene of interest, the expression level was normalized to the expression of the reference gene EEF1A2 (Eukaryotic translation elongation factor 1 alpha 2)[66]. Comparable results were obtained from SOL and EDL muscles of adult mice with EEF1A2 as the reference gene (Supplementary Fig. 10). Abbreviations: CSQ, calsequestrin; DHPRα$_{1S}$, dihydropyridine receptor α$_{1S}$ subunit; NCX, Na$^+$–Ca$^{2+}$ exchanger; Orai, Ca$^{2+}$ release-activated Ca$^{2+}$ channel; PMCA, plasma membrane Ca$^{2+}$-ATPase; RyR, ryanodine receptor; SERCA, sarco/endoplasmic reticulum Ca$^{2+}$-ATPase; STIM, stromal interaction molecule; TRPC, transient receptor potential cation channel. **b** Quantitative western blot analysis revealed no compensatory translational regulation ($P > 0.05$) of crucial proteins involved in EC coupling and Ca$^{2+}$ handling in the ncDHPR mouse ($n = 3$) compared with wt controls ($n = 3$). Band densities, quantified by densitometry, were standardized to GAPDH and normalized to the wt control sample on each gel. The corresponding western blots are shown in Supplementary Fig. 11 and in toto in Supplementary Fig. 12. Bars represent mean ± s.e.m. fold-change relative to wt; $P$ determined by unpaired Student's $t$-test

Although the hypothesis that the DHPR $Ca^{2+}$ current is vestigial seems strongly and clearly confirmed by all our findings, earlier studies on frog[25, 26], crab[27] and rat[28] skeletal muscles pointed to positive inotropic effects of different L-type $Ca^{2+}$ channel blockers. Recently, analogous inotropic effects were described by ex vivo $Ca^{2+}$ channel-blocking experiments on mouse diaphragm muscle[29]. It was hypothesized that DHPR $Ca^{2+}$ influx attenuates skeletal muscle contraction via inhibition of $Ca^{2+}$-sensitive AC isoforms, AC5 and AC6, which in turn leads to reduced intracellular cAMP levels, known to potentiate muscle contraction mainly through PKA-dependent RyR1 phosphorylation and thus preserving muscle fibre integrity[29]. With our ncDHPR mouse model we re-examined this apparently paradoxical $Ca^{2+}$ antagonist effect on muscle contraction. As demonstrated explicitly in the Results section, perfusion of isolated diaphragms of ncDHPR mice with two chemically distinct organic L-type $Ca^{2+}$ channel blockers, the DHP nifedipine and the phenylalkylamine (PAA) verapamil, surprisingly and in stark contrast to the hypothesis of Menezes-Rodrigues et al.[29], induced identical positive inotropic effects in wt and ncDHPR mice (Supplementary Fig. 13). With this experimental series, we could clearly exclude that $Ca^{2+}$ influx via the DHPR has a negative regulatory effect on skeletal muscle contraction and thus it is not essential for overall muscle health and fibre integrity as proposed in the earlier study[29]. However, our results cannot provide an explanation for this ($Ca^{2+}$-influx independent) apparently paradoxical $Ca^{2+}$ antagonist effect on muscle contraction. The most likely reason that we can speculate is that the antagonist concentrations (10 μM nifedipine and 50 μM verapamil) used in all these ex vivo experiments are ~ 6,000- and 25,000-fold, respectively, beyond the $K_d$-values observed for skeletal muscle membrane binding[54, 55] and thus might bind and interact non-specifically to low-affinity binding sites of other muscle proteins besides its specific high-affinity binding pocket on the $DHPR\alpha_{1S}$[56]. It is quite possible that such undefined low-affinity protein–DHP/PAA complexes are responsible for the observed amplification of skeletal muscle contraction.

Furthermore, our results and conclusions are also incongruent with a study on another mouse model (E1014K mouse) with a non-$Ca^{2+}$-conducting DHPR[31]. With this mouse model, it has been proposed that $Ca^{2+}$ permeation through (or just $Ca^{2+}$ binding to) the DHPR has a significant impact on muscle function. In contrast to the ncDHPR mouse model, the E1014K mouse displayed decreased SR $Ca^{2+}$ store refilling, increased fatigue, decreased muscle fibre size, increased type-IIb fibre count[31] and an obese phenotype[57]. The most probable reason for this disparity might be different biophysical consequences due to the implementation of distinct DHPR pore mutations. In the ncDHPR mouse, the N617D pore mutation adds a negative charge in close vicinity of the selectivity filter glutamate in repeat II of the $DHPR\alpha_{1S}$, which induces an aberrant high-affinity $Ca^{2+}$ binding to the pore[36]. This results in complete occlusion of the pore for all ions under physiological conditions. Conversely, in the E1014K mouse model, the E to K charge conversion in repeat III leads to impairment of the $Ca^{2+}$ selectivity filter, enabling only low-affinity (mM) $Ca^{2+}$ binding to the pore[32, 33]. Thus, the E1014K-DHPR does not allow $Ca^{2+}$ permeation but shows a strong conductivity for monovalent cations[31, 32], which consequently turns it into a junctionally targeted, slowly activating, non-inactivating $Na^+$ channel[34].

It is quite possible that this E1014K-specific leakiness for monovalent cations causes the observed alterations in muscle performance and is responsible for the differences between the two mouse models. Alternatively (or additionally), a proper conformation might be required for fine-tuning the DHPR signalling function, which is stabilized by high-affinity $Ca^{2+}$ pore-binding (as in the case of wt or N617D-DHPR) and destabilized by low-affinity $Ca^{2+}$ pore-binding (as in the case of E1014K-DHPR). Only this option allows discussing the claim of the study on the E1014K model that $Ca^{2+}$ permeation through and/or binding to the DHPR pore has an important modulatory role in muscle performance[31]. As the ncDHPR mouse does not show any of the aberrant muscle phenotypes found with the E1014K model, we can undoubtedly exclude the importance of DHPR $Ca^{2+}$ permeation for skeletal muscle performance. However, as $Ca^{2+}$ binds firmly to the N617D DHPR pore[36] in contrast to the E1014K channel, a putative crucial sole $Ca^{2+}$ binding effect to the DHPR to sustain skeletal muscle function cannot be excluded.

We are aware of the innate dilemma of a study, which intends to describe the non-existence of a phenotype and thus, despite an exhaustive scope of experiments, the fact that a stone was left unturned cannot be excluded. Nevertheless, all our results strongly suggest that the DHPR-mediated $Ca^{2+}$ current is fully dispensable in mammalian skeletal muscle and is most likely a non-functional, tolerated evolutionary remnant (vestigial $Ca^{2+}$ current) from the ancestral $Ca^{2+}$-influx-dependent skeletal muscle EC coupling found in early chordates[3, 7, 10]. The final switch-off of this vestigial skeletal muscle DHPR $Ca^{2+}$ influx took place only in the phylogenetically most derived clade among the vertebrates—the euteleost fishes[2, 3].

## Methods

**Generation of the *ncDHPR* knock-in mouse.** The strategy for transforming the mouse $DHPR\alpha_{1S}$ non conducting via introduction of point mutation N to D into exon 13, simultaneously with a silent mutation for restriction enzyme site PflF1 for RFLP test, was deliberated by our lab. Finally, the ncDHPR mouse on C57BL/6NTac background was generated at the Institut Clinique de la Souris—ICS (Illkirch, France).

Animal care and all experimental procedures were conducted in strict accordance with the guidelines of the European Union Directive 2010/63/EU and approved by the Austrian Ministry of Science (BMWF-5.031/0001-II/3b/2012). Mice were housed on a 12/12-h light/dark schedule, with access to food and water ad libitum. At least 24 h before the start of the experiments mice were housed individually and acclimatized to the behavioural facility. All experiments were performed on mutant and gender- and age-matched wt mice. Mice were organized into two age groups, "young" between 3 and 7 months and "aged" between 14 and 22 months old. Only male mice from both age groups were used in this study, unless indicated otherwise.

**Primary culture and whole-cell patch clamp on myotubes.** Myoblasts from skeletal muscles of new born up to 5 days old pups of wt, heterozygous or homozygous for non-conducting DHPRs were isolated and plated on 35-mm ECL-coated (Merck Millipore) culture dishes in Dulbecco's modified Eagle's medium in a 37 °C incubator with 5% $CO_2$[58]. L-type $Ca^{2+}$ currents simultaneously with intracellular SR $Ca^{2+}$ release using 0.2 mM Fluo-4 in the patch pipette solution were recorded from these cultured myotubes, as described[59]. Myosin-II blocker N-benzyl-p-toluene sulphonamide (BTS, 100 μM, Sigma) was continuously present in the bath solution.

**Two-electrode voltage clamp on isolated adult muscle fibres.** Measurements on voltage-clamped muscle fibres and signal analysis were performed as described[60]. Briefly, single muscle fibres were enzymatically isolated from interosseous muscles of young mice, voltage-clamped on a two-microelectrode setup and dialysed with an artificial intracellular solution containing 0.2 mM Fura-2 and 15 mM EGTA, to allow parallel fluorescence recordings of $Ca^{2+}$ currents and SR $Ca^{2+}$ release. The high intracellular EGTA concentration dominates $Ca^{2+}$ binding in these experiments and facilitates the calculation of the time course of the $Ca^{2+}$ release flux from the SR[61]. Depolarization-induced fluorescence ratio signals (380 and 360 nm excitation) were subjected to a removal analysis using a $Ca^{2+}$ distribution model fit approach[60]. The results were used to estimate the voltage-dependent $Ca^{2+}$ release flux and converted to SR $Ca^{2+}$ permeability by correcting for putative SR $Ca^{2+}$ depletion during the depolarizing voltage steps[60].

**SR $Ca^{2+}$ content measurements on isolated adult muscle fibres.** Primary-cultured interosseous muscle fibres[62] were loaded with 5 μM of the ratiometric low-affinity $Ca^{2+}$ indicator dye Fura-FF-AM in HEPES-buffered standard Krebs–Ringer solution for 30 min at room temperature, washed and equilibrated with Ringer solution containing 100 μM BTS for another 30 min to supress contractions. The bath solution was then replaced with $Ca^{2+}$-free Ringer solution to eliminate the

influence of external $Ca^{2+}$. Target cells were superfused using $Ca^{2+}$ releasing solution, i.e., $Ca^{2+}$-free Ringer solution containing 500 µM of the RyR agonist 4-CmC to trigger $Ca^{2+}$ release and 30 µM CPA to inhibit SERCA $Ca^{2+}$ sequestration. A period of 20-s application of the $Ca^{2+}$ releasing solution was always bracketed with 20-s superfusion with $Ca^{2+}$-free Ringer solution (Fig. 3a). For $Ca^{2+}$ measurements, the ratio of fluorescence at 340 and 380-nm excitation was determined using the Zeiss Microscope Photometer System (FFP) based on an inverted microscope (Axiovert 35, Zeiss) equipped for epifluorescence (Fluar ×40/1.3 oil-immersion objective)[63].

**Fibre-type and fibre CSA analysis.** Eight-micrometre-thick cryosections of snap-frozen SOL and EDL muscles from 2 to 3 months old ncDHPR and wt mice were immunostained overnight at 4 °C with antibodies against fibre-type-specific myosin heavy chain (MHC) isoforms. Primary antibodies against MHC isoforms I (BS-D5), IIa (SC-71), IIb (BF-F3) and IIx (6H1) (Developmental Studies Hybridoma Bank, USA) at a dilution of 1:2,000, except 6H1, which was 1:200, were used. Secondary staining was performed for 1 h at room temperature (RT) with anti-mouse IgG Alexa 594 or anti-mouse IgM Alexa 594 (1:4,000, Invitrogen). Images were taken under a fluorescence microscope (Olympus BX53) with a ×20 objective lens. The number of antibody-stained fibres were counted and normalized to the total number of muscle fibres per field[64]. Saved images were additionally used for calculating the fibre CSA with Metamorph software (Molecular Devices, USA).

**Ex vivo force frequency and fatigue measurements.** Intact muscle-tendon complexes of EDL and SOL were dissected from the hindlimbs of euthanised young or aged wt or ncDHPR mice[65] and mounted vertically between a force transducer (Model FT03, Grass Instruments, Quincy, USA) and static clamp in an organ bath equipped with platinum electrodes and under continuous perfusion with carbogen- (95% $O_2$ + 5% $CO_2$) saturated Krebs–Ringer solution. Optimal muscle stretch was determined by applying single twitches at supramaximal voltage (25 V for 1 ms) and set at the length that generated maximal force. After 10 min of equilibration, EDL and SOL muscles were subjected to different force frequency (tetani with increasing stimulation frequencies but fixed 2-min recovery intervals) and repetitive tetanic fatiguing (decreasing recovery intervals by 27% every 2 min) protocols[38] (Supplementary Fig. 7). All experiments were performed at room temperature (~26 °C). Data acquisition and analysis was carried out using custom made software (Delphi, Borland). Muscle length, diameter and wet weight were measured at the end of each experiment.

**Voluntary locomotor activity measurements.** Spontaneous physical activity of 3–6 months old male mice was measured in their home-cage environment with free access to food and water. Mice were singularized at least 1 week before the start of the experiment.

Spontaneous total activity was monitored by a home cage ambulatory activity test over a period of 2 days and 3 nights with an infrared sensor unit (InfraMot, TSE Systems, Homburg, Germany) mounted on top of every home cage.

For voluntary activity wheel test, mice were individually habituated for 2 days before the start of the experiment in cages furnished with activity wheels (Bioseb, Vitrolles, France). The mice had continuous free access to the wheels, which could be revolved in either direction. Activity wheels were equipped with photoelectric sensors interfaced to a computerized system that automatically recorded the distance travelled every 5 min continuously for 21 days. Up to six mice were run in parallel and for each genotype the distances travelled per day were calculated and plotted as averaged, cumulative values.

**Muscle coordination, strength and endurance measurements.** Three standard tests were used to compare the motor coordination and muscle strength between ncDHPR and wt mice.

For the beam walk test, ncDHPR or wt mice from the young group were used. The mouse was placed on one end of a 120-cm long beam resting 50 cm above a padded surface. An enclosed safety box was mounted at the other end of the beam. After three pre-training trails, the time needed to traverse the beam and reach the safety box was recorded for every mouse.

For the wire hang test, a custom-designed device consisting of a cross-wired grid, positioned between two poles 35 cm apart and 55 cm above a padded surface was constructed. The mouse was placed in the centre of the grid and shaken gently three times, to ensure firm gripping to the wire (1.7 mm in diameter) with all four paws. The grid was then immediately turned upside down and the latency to fall was recorded. Three trials were done with every mouse, with a recovery interval of some minutes, and only the highest score was used for analysis. The maximum hanging time was set to 600 s and the mouse that did not fall during this period was removed and given the maximum score. Body weight was recorded at the end of the experiments and the average weight was similar between ncDHPR and wt mice.

Maximal muscle strength of the forelimbs was measured using a grip strength test (Bioseb). The mouse was gently lowered by the tail and allowed to grip a metal bar (1.3 mm in diameter) with the front paws only. Keeping the torso horizontal, a slow gradual pull was exerted in backward direction until the grip was released. The force applied to the bar by the animal at the time of the release is recorded by the sensor as the maximum force reached. Three measurements separated by 30-s

inter-trial intervals were collected from each animal and the maximum value was used for statistical analysis. Mice were weighed at the end of the testing session and there was no difference in average weight between ncDHPR and wt mice.

**Acute fatigue measurements.** Two standard methods were used to compare fatigability and endurance between ncDHPR and wt mice.

Rotarod tests at accelerating and constant speeds were used to assess muscle fatigue and endurance. Mice were familiarized and trained on a four-lane rotarod (TSE Systems) by running them for 5 min per day for 3 days at a speed of 4 rpm. For testing the fatigability, young and aged ncDHPR and wt mice were placed on the rotarod at an initial speed of 4 rpm for 25 s. After the warmup phase, the rotarod was set to accelerate steadily from 4 to 40 rpm for 300 s. Each mouse was subjected to three trials with 15 min inter-trial intervals. For rotarod endurance task, mice were placed on the rod rotating at a constant speed, 20 rpm for young and 15 rpm for aged mice for 600 s. Latency to fall was recorded and only the highest score among the 3 trails was considered for analysis.

Treadmill test was performed to assess the degree of fatigability under rigorous conditions. Mice were run on a five-lane treadmill with electric stimulus grid (Bioseb). Three days before the experiment, mice were acclimatized and trained on the treadmill by running them for 5 min per day at a speed of 9 cm s$^{-1}$ at a 15° inclination. Acute exhaustion protocols following an initial 5-min warm up at 5 cm min$^{-1}$ were designed for young and aged mice. In case of young mice, band speed was increased 2 cm s$^{-1}$ every minute till 27 cm s$^{-1}$ and then maintained at the maximum speed for an additional 45 min. The entire treadmill task was 0.9 km in 1 h. For aged mice, a milder protocol with a speed increment of 1 cm s$^{-1}$ every minute up to 17 cm s$^{-1}$ was used. The entire run was 0.55 km distance in 1 h. To encourage continuous forward running, the front end of the treadmill was covered with a black cloth and electric stimulus at the back end was set at 0.2 mA. The number of rests and body weight was recorded for each mouse. Finally, the average cumulative resting time collapsed into 5 min bins, was plotted against the duration of the task. In addition, average number of cumulative rests during the entire task was calculated.

**RNA isolation and qPCR.** Putative transcriptional regulation of key triadic proteins involved in EC coupling and $Ca^{2+}$ homeostasis was assessed by comparative TaqMan quantitative reverse-transcriptase PCR (qRT-PCR). Total RNA was extracted from skeletal muscle of neonatal pups (up to 2 days old) and adult (7 months old) EDL and SOL muscles using RNeasy Mini Kit (Qiagen). Following RNAse-free DNAse treatment, first-strand complementary DNA synthesis was performed using random primers and M-MLV reverse transcriptase (Promega). For every gene, flanking primer pairs and 5′ FAM (6-carboxyfluorescein) and 3′ BHQ1 (Black Hole Quencher)-labeled TaqMan probe spanning exon–exon boundaries, were designed and purchased from Eurofins, Germany (Supplementary Table 2). qRT-PCR assays were performed in triplicates with GoTaq Probe qPCR master mix (Promega) on a Real-Time thermal cycler (Montania 483, Anatolia Geneworks, Turkey). Relative expression levels were calculated using the comparative $C_T$ method with two endogenous reference genes—ACTB (β-actin) and one of the most stable mammalian skeletal muscle housekeeping gene, EEF1A2 (eukaryotic translation elongation factor 1 alpha 2)[66].

**Western blot analysis.** TA muscles from 2 to 3 months old ncDHPR and wt mice were excised, homogenized and lysed in sample buffer containing 250 mM Tris-HCl (pH 8.5), 2% lithium dodecyl sulfate, 10% glycerol and 0.5 mM EDTA and centrifuged at RT, 16,000×g for 5 min. Protein concentration measurements and separation by SDS-polyacrylamide gel electrophoresis on 3–8% Tris-Acetate or 4–12% Bis-Tris NuPAGE gels (Thermo Fisher Scientific) was performed as described previously[67]. Immunostaining was performed with different primary antibodies followed by appropriate secondary antibodies (Supplementary Table 3). Immunoreactive bands were visualized on the Odyssey scanner (LI-COR). Western blot densitometries were performed using Image Studio Lite software, with GAPDH as a loading control. Uncropped images of western blots are shown in Supplementary Fig. 12.

**Ex vivo isometric contraction recordings on diaphragm muscle.** Three- to 5-mm strips with rib overhangs were dissected from diaphragm muscle isolated from euthanized 3–6 months old ncDHPR or wt mice. Strips were mounted vertically in the ex vivo contraction setup (as described above) with the central tendon facing up and the rib end towards the bottom and kept under continuous perfusion with carbogen (95% $O_2$ + 5% $CO_2$) saturated Krebs–Ringer solution. Optimal muscle stretch was determined by applying a single twitch at supramaximal voltage (25 V for 2 ms). After 10 min of equilibration under continuous stimulation (0.33 Hz, 2 ms, 25 V), nifedipine (10 µM) or verapamil (50 µM) was added to the bath solution to examine the effect of L-type $Ca^{2+}$ channel blockers on muscle contraction[29]. Nifedipine was washed out after 25 min and stimulation continued for another 15 min, whereas verapamil was washed out after 40 min with further stimulation for 20 min. At the end of each experiment, muscle wet weight, diameter and length were measured. All recordings were performed at room temperature (~26 °C). Baseline force was calculated by taking the average twitch peak force of the initial 6 min before adding the drug. Finally, the average peak twitch force,

collapsed into 2-min bins and expressed as percentage of baseline force, was plotted against the duration of the experiment.

**General experimental design and statistical analyses**. Sample sizes of mice or tissue are based on previous publications[2, 29, 31, 38, 39, 64]; hence, power calculations were not necessary. Animals were arbitrarily allocated to the specific groups (according to genotype, sex and age), but without the use of an explicit randomization procedure. Experiments did not require blinding and thus were not performed under blinded conditions. In our study, no data points, samples or mice were excluded from analysis. Based on our previous publications[2, 3, 35, 44, 45, 52, 56, 59], all statistical analyses are considered appropriate. $n$-Values represent the number of mice, muscles, muscle fibres or myotubes, as specified. Variance is similar between comparison groups. All data are expressed as means ± s.e.m. and statistical analyses were conducted using unpaired Student's $t$-test. $P < 0.05$ was considered statistically significant.

**Data availability**. All relevant data are available from the first/corresponding author on reasonable request.

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

## Acknowledgements

We thank Dr Christoph Schwarzer for providing support and access to the Home cage and Rotarod setup, Dr Gerald Obermair for help with the qPCR analysis, Drs Zoran Culig and Florian Handle for helping with western blot analysis, Dr Bernhard Flucher and Stefanie Geisler for providing support in examining the muscle fibre-type composition, Dr Veit Flockerzi for providing the anti-Orai1 antibody, Drs Robert T. Dirksen and Lan Wai-LaPierre for most of the other WB antibodies and Achim Riecker for help with the contraction setup. We also thank Birgit Kagerbauer and Karin Fuchs for excellent technical assistance. This study was supported by the Austrian Science Fund (FWF) Grants (P23229, P27392 and DK-W1101 to M.G.) and W.M. received funding from the Deutsche Forschungsgemeinschaft (DFG) Grant (ME-713/18).

## Author contributions

A.D. and M.G. designed research; Y.P. performed electrophysiology on muscle fibres and together with K.F. the SR $Ca^{2+}$ loading experiments; A.D., M.G. and K.S., W.M. performed all the other experiments; and A.D. and M.G. wrote the manuscript.

## Additional information

**Competing interests:** The authors declare no competing financial interests.

