## [Peer Review file · Nature Communications]

Reviewers' comments:

Reviewer #1 (Remarks to the Author):

The physiological role, if any, of L-type Ca^{2+} current in mammalian skeletal muscle of has remained enigmatic for nearly half a century, particularly since this flux is not required to trigger the SR Ca^{2+} release that occurs during excitation-contraction (EC) coupling. The Grabner group has demonstrated elegantly that, through the ages, teleosts (e.g., zebrafish) have shed the burden of the L-type Ca^{2+} current through the introduction of residues in their $\text{Ca}_v1.1$ isoforms that render the channels non-conducting. To test the importance of L-type Ca^{2+} into mammalian skeletal muscle, Dayal and colleagues have created a mouse line in which one of these zebrafish residues (N617D) has been introduced into the mouse $\text{Ca}_v1.1$ channel. These mice were vigorous, fertile and generally indistinguishable from wild-type littermates. As expected, the N617D swap ablated L-type Ca^{2+} current without effect on the channel's ability to serve as voltage-sensor for EC coupling in both cultured myotubes and acutely dissociated *interosseous* fibres obtained from homozygous mice. Further, the authors found virtually no effect of ablation of Ca^{2+} influx via $\text{Ca}_v1.1$ on SR Ca^{2+} store content, body weight, muscle size/mass, twitch or tetanic force, muscle fatigue, locomotor function and the transcript levels of $\text{Ca}_v1.1$ and other established mediators of Ca^{2+} signaling in skeletal muscle (e.g., RyR1, TRPC1, Orai1, STIM1, SERCA1, etc.). These observations led the authors to the conclusion that L-type Ca^{2+} flux is not required for normal skeletal muscle function in mammals and that it is therefore vestigial. This conclusion is controversial since another group has found impaired EC coupling, altered muscle composition, force deficits and metabolic issues with a mouse strain carrying a distinct non-conducting $\text{Ca}_v1.1$ mutant channel (E1014K; Lee et al., 2015).

The authors are meticulous and comprehensive in their tests. Based on the nearly exhaustive scope of these experiments, their overall conclusion is reasonable. However, the caveat must be made that they may have left a stone unturned; the L-type current may have a function at some level, but that particular function was not tested. Another caveat that probably should be addressed is that transcript levels of the major players in EC coupling don't necessarily equate to their activity levels since a number of these proteins are regulated by post-translational processes including, but not limited to, phosphorylation, oxidation/reduction, etc. Still, the evidence towards the conclusion that the L-type current is irrelevant for normal skeletal muscle function in mammals is strong.

The one thing that seems to be missing from the assessment of the impact of L-type current ablation is an examination of muscle fiber morphology and fiber-type composition. Even though the *ex vivo* contractile experiments argue against alterations in muscle composition, such information seems quite important considering recent findings of Lee et al. (2015) and Sultana et al. (2016) which indicated that either ablation or amplification, respectively, of L-type current can profoundly impact fiber-type content.

The comparison of the non-conductance mechanisms for E1014K and N617D needs to be modified as both channels display Ca^{2+} block (see Fig. 2 of Bannister and Beam, 2011). Unless I'm missing something here, the former mutation produces block through stabilization of Ca^{2+} binding within the selectivity filter by ablating of one of $\text{Ca}_v1.1$'s two Ca^{2+} binding sites, whereas the latter seems to stabilize Ca^{2+} (or TEA⁺ or Na⁺, for that matter) binding by adding the extra negative charge in the immediate vicinity of the filter. These contrasting mechanisms don't negate the authors' sub-conclusion that differential Ca^{2+} binding within the selectivity filter could account for the phenotypic differences between E1014K and N617D mice, but the mechanisms should be more clearly articulated.

The last sentence of the Discussion is cryptic. Please elaborate or delete.

Reviewer #2 (Remarks to the Author):

The authors have addressed an age-old and intriguing question regarding the utility of Ca²⁺ influx through the voltage-gated Ca²⁺ channel in skeletal muscle (i.e., CaV1.1). The authors are experienced in genetic and electrophysiological techniques needed to generate a novel homozygous knock-in mouse model that lacks Ca²⁺ influx via CaV1.1 and assess Ca²⁺ handling in muscle fibers from this model. The authors also employ a diverse battery of functional tests to assess the impact of the knock-in model on function at the muscle to whole animal levels. The authors demonstrate that homozygous knock-in mice that lack CaV1.1 Ca²⁺ influx exhibit no differences in SR Ca²⁺ release, locomotor activity, motor-coordination, muscle strength, and susceptibility to fatigue compared to wild type mice. The authors conclude that their findings support the notion that the CaV1.1-mediated Ca²⁺ influx in mammalian skeletal muscle is an evolutionary remnant.

Although most of the results are compelling and do support their hypothesis, the study would be strengthened if it addresses a critical characteristic of skeletal muscle: its ability to adapt to exercise stimuli (e.g., strength and endurance training). Contractile and metabolic phenotype plasticity is a hallmark of skeletal muscle and should be addressed before concluding that Ca²⁺ influx serves absolutely no purpose. Given the role of DHPR in activating the G proteins-PI3 kinase-PLC-IP3 signaling cascade (e.g., Casas et al., *J. Gen Physiol* 136: 455-467, 2010) and the putative role of DHPR and IP3 in transducing the frequency signals of exercise training (Jorquera et al., *J. Cell Sci* 126: 1189-1198, 2013), the strength of the argument supporting the hypothesis would be greatly enhanced by addressing these issues before drawing their final conclusion. In addition, the EK mouse model that lacks CaV1.1 Ca²⁺ influx does exhibit significant effects on muscle size, fiber type, protein metabolism, SR Ca²⁺ stores, fatigue, and age-related strength mediated in large part via CaMKII signaling (Lee et al. *Skeletal Muscle* 5:4 DOI 10.1186/s13395-014-0027-1, 2015). Although the authors address some of these findings in their Discussion, the current study would be strengthened if they address in more detail basic skeletal muscle phenotype (i.e., fiber type) and signaling pathways known to influence to phenotype. Moreover, the authors should consider adding the potential role of Ca²⁺ flux through the DHPR in affecting skeletal muscle phenotype in their Introduction (51-54).

The battery of functional tests is impressive, however the authors should provide references that indicate that they are accurate and reliable measures of what they report to measure, especially given the extremely low statistical power values (e.g., 0.05 and 0.13 for mean latency to fall during 600-s Rotarod endurance task at constant speed) on a number of the tests. In addition, critical measures of isolated muscle function (i.e., twitch and tetanic forces) appear to have been made at relatively low temperatures based on the shape of the force-frequency relationship. Please provide the temperatures at which the physiological experiments were conducted (i.e., Methods or Figure legends), and if the measurements were made at relatively low temperatures, the authors should provide a rationale and address any potential impact on their conclusions.

Considering that 10 out of 12 genes of interest in the ncDHPR mice exhibited an approximate 10-20% lower expression than WT mice, the study would be greatly enhanced if the corresponding protein levels were also determined in view of their direct role in EC coupling and potential to influence cell signaling.

Suggested minor comments to enhance readability of paper:

(49-51) The following sentence is somewhat awkward and should be revised for clarity: Even though, the physiological role of this enigmatic Ca²⁺ influx which is not (immediately) required for

EC coupling is under debate since nearly half a century, a convincing answer elucidating its function is still in the dark.

Please provide the actual p values instead of directional p values (i.e., $p > 0.05$). There are several cases where genotype differences are relatively large, e.g., line 174, ncDHPR exhibited 53% greater wire hang strength than WT (p value is 0.071 with a power of 0.32). Therefore, with a low statistical power (accepted values are typically 0.80) and a p value close to 0.05, drawing a definitive conclusion that there is no difference with this result is tenuous.

(106-108) Please indicate that voltage-induced SR Ca²⁺ transients were also normal in the following sentence: Altogether, these results indicate a complete elimination of skeletal muscle Ca²⁺ influx in the ncDHPR mouse model and thus serves as a unique system for investigating the physiological relevance of the DHPR Ca²⁺ influx in skeletal muscle EC coupling.

(107-125) The use of the word "initial" seems odd given that the data describes adult animals, and the word "phenotype" leaves the reader expecting more information about the mouse other than just its body weight and litter size. Please consider revising and combining this data with measurements of skeletal muscles described later and ideally fiber type composition.

(126-128) Suggest revising to read "Since the ncDHPR mouse did not show any evident phenotype differences from wt,..."

(128) Suggest revising to read "Home-cage ambulatory activity recorded..."

(132-133) Please add the average age of the mice.

(134, 150) Replace "identical" with "similar" (the values are not identical).

(140) Mouse soleus muscle is more "slow-twitch" than mouse EDL, however it is not predominately "slow" from a traditional fiber-type classification based on MHC isoforms (e.g., Pelligrino et al Eur J Appl Physiol. 2005 Mar;93(5-6):655-64).

(142) Please provide the average age of the "aged" mice (14 months of age is not considered old for these mice).

(174) This functional test is a measure of muscular endurance and not maximal muscular strength.

(190-191) Please revise the following "in the absence of extracellular Ca²⁺ influx in ncDHPR mice" because the authors did not measure total Ca²⁺ flux into muscle fibers nor through any other Ca²⁺ channels (e.g., SOCE, SAC).

(229-230) The authors state that "...either influx or efflux of Ca²⁺ ions compensates for the reduced myoplasmic Ca²⁺ concentration in ncDHPR mice." However, the authors should revise this statement since they only measured Ca²⁺ flux through ncDHPR and not through other Ca²⁺ channels or the [Ca²⁺]_i after contractile activity.

(268) Please add "force" in the following: ...robust increment in muscle twitch contraction force in wt mice.

Respectfully,
Christopher P. Ingalls, Ph.D.

Reviewer #3 (Remarks to the Author):

This is a clearly developed and articulated study that provides comprehensive and clear-cut tests with respect to the somewhat controversial functional role of Ca influx via the L-type Ca channel (LTCC) in mammalian skeletal muscle. The authors have used a knock-in mouse that expresses an LTCC mutant that completely prevents current via the Ca channel. This mutation also prevents monovalent cation current via the LTCC, which is a potential limitation for an excellent 2015 knock-in mouse study of similar overall strategy (ref 53). In my mind this manuscript provides definitive evidence that the residual Ca current that is carried by the normal mouse skeletal muscle LTCC (Cav1.1) is vestigial, and that its loss has no discernable functional consequence. The work is comprehensive from biophysical characterizations to integrated function, behavior and aging. This study is a major advance for this field. My only comments are for minor stylistic and clarifying points.

1. The related 2015 paper from Hamilton's group should probably be mentioned in the Introduction section (not just at the end of the Discussion). That prior study used the same "overall" strategy, and the functional distinction between the 'EK-mutant' and the N617D mutant used here is important in both justifying the novelty of this study, and the implications of the results. That is, it can set up the present study in an even stronger way, and I would say that this could be done without lengthening the Introduction.

2. Related to that manuscript length point, the overall style is somewhat more wordy than necessary. The Intro could be a bit more concise, without loss of content and the Results which repeat nearly every number that is already displayed in the figures (and sometimes to 5 significant(?) figures). I suggest that reciting the numbers be reserved for a few key points that the authors want to make (even if they prefer to put the actual values in the legends. I do understand their desire to be comprehensive in that way, but it does break up the flow of the text.

Other minor style points that made reading of this otherwise very clear manuscript more difficult as a reviewer was the lack of clear paragraph breaks (by indents or extra spacing) and lack of numbers on the Figure pages. Obviously these are issues the copy editing will remedy, so this is mainly advice for future.

3. On line 229 it is implied that there is reduced myoplasmic Ca in the ncDHPR mice, but no differences in Ca transients, Ca content or resting $[Ca]_i$ measurements were reported. I think you mean "...for the reduced Ca influx via DHPR in the ncDHPR mice." or "...for the possible reduced myoplasmic Ca levels that might have been anticipated in the ncDHPR mice."

In this same spot (ln 230 & Fig 9), the graphs should indicate that these were mRNA measurements, so that one immediately knows upon looking at the Fig that this is message, not protein. Likewise on line 239, change to "Furthermore, expression profiles of mRNA that codes for critical Ca...".

Minor edits:

Line 270; replace "as well" with "also"

Line 276; Waning of the response was not apparent during the 25 min exposure period, so I suggest deleting "was transient and the effect".

Line 289; replace "recent function" with "functional importance"

Line 307; delete "this mysterious"

Our point-by-point response to the reviewers:

We would like to thank the reviewers for their overall positive attitude and constructive comments which will certainly improve our manuscript. Accordingly, we have revised the manuscript and implemented suggested changes and additions. All changes in the manuscript text file are highlighted in blue. Detailed answers to the reviewers' comments are as follows:

Our answers to the comments of **Reviewer #1**.

The physiological role, if any, of L-type Ca^{2+} current in mammalian skeletal muscle of has remained enigmatic for nearly half a century, particularly since this flux is not required to trigger the SR Ca^{2+} release that occurs during excitation-contraction (EC) coupling. The Grabner group has demonstrated elegantly that, through the ages, teleosts (e.g., zebrafish) have shed the burden of the L-type Ca^{2+} current through the introduction of residues in their CaV1.1 isoforms that render the channels non-conducting. To test the importance of L-type Ca^{2+} into mammalian skeletal muscle, Dayal and colleagues have created a mouse line in which one of these zebrafish residues (N617D) has been introduced into the mouse CaV1.1 channel. These mice were vigorous, fertile and generally indistinguishable from wild-type littermates. As expected, the N617D swap ablated L-type Ca^{2+} current without effect on the channel's ability to serve as voltage-sensor for EC coupling in both cultured myotubes and acutely dissociated interosseous fibres obtained from homozygous mice. Further, the authors found virtually no effect of ablation of Ca^{2+} influx via CaV1.1 on SR Ca^{2+} store content, body weight, muscle size/mass, twitch or tetanic force, muscle fatigue, locomotor function and the transcript levels of CaV1.1 and other established mediators of Ca^{2+} signaling in skeletal muscle (e.g., RyR1, TRPC1, Orai1, STIM1, SERCA1, etc.). These observations led the authors to the conclusion that L-type Ca^{2+} flux is not required for normal skeletal muscle function in mammals and that it is therefore vestigial. This conclusion is controversial since another group has found impaired EC coupling, altered muscle composition, force deficits and metabolic issues with a mouse strain carrying a distinct non-conducting CaV1.1 mutant channel (E1014K; Lee et al., 2015).

The authors are meticulous and comprehensive in their tests. Based on the nearly exhaustive scope of these experiments, their overall conclusion is reasonable. However, the caveat must be made that they may have left a stone unturned; the L-type current may have a function at some level, but that particular function was not tested.

We fully agree with the reviewer that from a theoretical / philosophical point of view the “absence of evidence doesn't inevitably imply evidence of absence”, especially if we look at a more or less “open system”, like an organism influenced by environmental stimuli. Thus, it is correct that despite our “nearly exhaustive scope of experiments” we theoretically might have missed a particular function where “the L-type current may have a function at some level”

To make a point, a sentence has been added to the Discussion section (lines 404-407): “We are aware of the innate dilemma of a study which intends to describe the non-existence of a phenotype and thus,

despite an exhaustive scope of experiments, the fact that a stone was left unturned cannot be excluded. Nevertheless, all our results strongly suggest that the DHPR-mediated Ca²⁺ current is fully dispensable in mammalian skeletal muscle and.....”.

Another caveat that probably should be addressed is that transcript levels of the major players in EC coupling don't necessarily equate to their activity levels since a number of these proteins are regulated by post-translational processes including, but not limited to, phosphorylation, oxidation/reduction, etc. Still, the evidence towards the conclusion that the L-type current is irrelevant for normal skeletal muscle function in mammals is strong.

The reviewer is correct that transcript levels of the major players in EC coupling do not necessarily equate to their activity levels. Therefore, and also in accordance to the wish of reviewer #2, we performed quantitative western blot analysis of key triadic EC coupling proteins with the aim to evaluate putative changes also at the protein level. Results are added in the revised manuscript as Fig. 9b, Supplementary Fig. S11 and S12 and in the Result section; lines 252-258, as well as briefly summed up in the Discussion section (lines 338-340). In addition, we added the following statement in the Discussion section (lines 344-347): *“However, analyses of mRNA or protein levels of these major players in EC coupling and Ca²⁺ homeostasis do not necessarily equate to their activity levels since a number of these proteins are post-translationally regulated. Still, the evidence that these key proteins remain uninfluenced by the lack of DHPR Ca²⁺ influx is very high.”*

The one thing that seems to be missing from the assessment of the impact of L-type current ablation is an examination of muscle fiber morphology and fiber-type composition. Even though the ex vivo contractile experiments argue against alterations in muscle composition, such information seems quite important considering recent findings of Lee et al. (2015) and Sultana et al. (2016) which indicated that either ablation or amplification, respectively, of L-type current can profoundly impact fiber-type content.

We agree with the reviewer that muscle fibre morphology and fibre-type composition might be essential in the light of the cited studies, irrespective of the fact that we did not find any muscle phenotype in any of our experiments at *ex vivo* and whole animal levels. Therefore, and also in accordance to the suggestion of reviewer #2, we implemented muscle fibre cross-sectional area (CSA) and fibre-type analyses in our revised manuscript. Results are added in the Supplementary section as Fig. S6 and in the Result section; lines 139-144. In addition, the following note was added to the Discussion section (lines 331-333): *“..... lack of DHPR Ca²⁺ influx in ncDHPR mice is inconsequential for body weight development, fertility, locomotor activity, muscle fibre-type composition, fibre CSA, muscle strength and susceptibility to fatigue”*

The comparison of the non-conductance mechanisms for E1014K and N617D needs to be modified as both channels display Ca²⁺ block (see Fig. 2 of Bannister and Beam, 2011). Unless I'm missing something here, the former mutation produces block through stabilization of Ca²⁺ binding within the selectivity filter by

ablating of one of Ca_v1.1's two Ca²⁺ binding sites, whereas the latter seems to stabilize Ca²⁺ (or TEA⁺ or Na⁺, for that matter) binding by adding the extra negative charge in the immediate vicinity of the filter. These contrasting mechanisms don't negate the authors' sub-conclusion that differential Ca²⁺ binding within the selectivity filter could account for the phenotypic differences between E1014K and N617D mice, but the mechanisms should be more clearly articulated.

Our comparison of the non-conductance mechanisms for E1014K and N617D is based on a widely accepted model of Ca²⁺ channel permeation and selectivity, based on the two exciting papers from Richard Tsien's lab (Yang et al., *Nature*, 1993; Ellinor et al., *Neuron*, 1995), which were later comprehensively discussed in the review of Sather & McCleskey, (*Annu Rev Physiol.*; 2003). According to this model, Ca²⁺ binds to a single high-affinity site formed by all 4 glutamates of the selectivity filter of Ca_v1 and this tight embracement of Ca²⁺ is a prerequisite for the high selectivity of the Ca_v1 pore for Ca²⁺ over Na⁺ or other monovalent cations. However, to allow rapid passage of Ca²⁺ through the pore, a two-site mechanism is essential that overcomes this tight Ca²⁺ binding. Accordingly, the EEEE locus has been suggested to be physically flexible. Despite all 4 selectivity filter glutamates are needed to hold a single Ca²⁺ ion with high affinity ($K_D \sim 1 \mu\text{M}$), their conformation can rapidly rearrange to accommodate a pair of Ca²⁺ ions within the pore, but now both bound with much lower affinity (apparent $K_D \sim 14 \text{ mM}$). This short-lived low-affinity intermediate stage, together with a mechanism of Ca²⁺-Ca²⁺ repulsion occurring in this doubly occupied pore, is the basis of fast Ca²⁺ ion passage through the pore, pushing one of the occupying Ca²⁺ ions out to the cytosolic side.

The strongest impact on the binding affinity (measured as Ca²⁺-block of Li⁺ currents through the Ca_v1-pore) was produced by exchanges of E in repeat III. EIIIK mutations reduced the pore's affinity for Ca²⁺ as high as 1000-fold – raising the IC₅₀ for Ca²⁺ block of I_{Li} from $\sim 1 \mu\text{M}$ to $\sim 1 \text{ mM}$. (Yang et al., *Nature*, 1993).

Although the experiments described above were performed in Ca_v1.2, the selectivity/conductance model seems also to be congruent with Ca_v1.1 experiments. Accordingly, the large Cs⁺ outward current found in the Ca_v1.1 EIIIK mutant (SkEIIIK) which was not blocked even in the presence of 10 mM external Ca²⁺, was consequently interpreted as an indication of very little residual binding of Ca²⁺ within the SkEIIIK pore (Dirksen & Beam, *J. Gen. Physiol.*, 1999). Similarly, a considerable Na⁺ current through SkEIIIK can be found despite 10 mM of external Ca²⁺ (see Fig. 2: Bannister & Beam, *Channels*, 2011). This again indicates a mere marginal, low-affinity binding of Ca²⁺ within the SkEIIIK pore, which abolishes Ca²⁺ conductance as well as Ca²⁺ selectivity.

We interpret the blocking mechanism in N617D exactly like stated by the reviewer. As shown in panel 6 of the attached poster (Dayal & Grabner, *paper in preparation*), the additional negative charge introduced due to D617 in close vicinity to the selectivity filter EII, induces an aberrant high-affinity Ca²⁺ binding to the pore. Our results indicate a shift of IC₅₀ for Ca²⁺ block of I_{Li} from $\sim 1 \mu\text{M}$ to the nM range, resulting in a full occlusion of the pore for all ions under physiological conditions.

In the revised manuscript, we discussed the comparison of the non-conductance mechanisms for

both E1014K and N617D in a clearer way and cited the review of Sather & McCleskey (2003) at appropriate positions (lines 60-68 of the Introduction and lines 377-403 of the Discussion).

The last sentence of the Discussion is cryptic. Please elaborate or delete.

As suggested by the reviewer, the last sentence of the discussion has been deleted.

Our answers to the comments of **Reviewer #2**.

The authors have addressed an age-old and intriguing question regarding the utility of Ca²⁺ influx through the voltage-gated Ca²⁺ channel in skeletal muscle (i.e., CaV1.1). The authors are experienced in genetic and electrophysiological techniques needed to generate a novel homozygous knock-in mouse model that lacks Ca²⁺ influx via CaV1.1 and assess Ca²⁺ handling in muscle fibers from this model. The authors also employ a diverse battery of functional tests to assess the impact of the knock-in model on function at the muscle to whole animal levels. The authors demonstrate that homozygous knock-in mice that lack CaV1.1 Ca²⁺ influx exhibit no differences in SR Ca²⁺ release, locomotor activity, motor-coordination, muscle strength, and susceptibility to fatigue compared to wild type mice. The authors conclude that their findings support the notion that the CaV1.1-mediated Ca²⁺ influx in mammalian skeletal muscle is an evolutionary remnant.

Although most of the results are compelling and do support their hypothesis, the study would be strengthened if it addresses a critical characteristic of skeletal muscle: its ability to adapt to exercise stimuli (e.g., strength and endurance training). Contractile and metabolic phenotype plasticity is a hallmark of skeletal muscle and should be addressed before concluding that Ca²⁺ influx serves absolutely no purpose. Given the role of DHPR in activating the G proteins-PI3 kinase-PLC-IP3 signaling cascade (e.g., Casas et al., *J. Gen Physiol* 136: 455-467, 2010) and the putative role of DHPR and IP3 in transducing the frequency signals of exercise training (Jorquera et al., *J. Cell Sci* 126: 1189-1198, 2013), the strength of the argument supporting the hypothesis would be greatly enhanced by addressing these issues before drawing their final conclusion.

We have to respectfully disagree with the reviewer in this specific point. I know both the cited papers from Enrique Jaimovich's lab very well (2nd paper being a follow-up of the 1st paper). I had the opportunity to discuss extensively the described phenomenon of stimulation-dependent slow Ca²⁺ waves, shown to induce excitation-transcription (E-T) coupling and thus muscle plasticity, from different perspectives with him.

In a nutshell: It is an irrevocable fact that activation of the specific transcriptional programs that define the muscle cells' phenotype involve the DHPR (Ca_v1.1) ONLY as a voltage sensor (to activate the above described cascade by conformational protein-protein interaction) and not in its function as Ca²⁺-conducting channel. Corresponding statements about the independence of this mechanism from DHPR Ca²⁺-influx (Jorquera et al., *J. Cell Sci* 126: 1189-1198, 2013) can be found in the 2nd last line of the Abstract, as last statement in the Results, in the scheme in Fig. 7, and very explicit in the 5th

paragraph of the Discussion: “We have previously shown that *Ins(1,4,5)P3*-dependent Ca^{2+} signals are independent from extracellular Ca^{2+} entry (Casas et al., 2010). The fact that nifedipine and BayK have the same inhibitory effects on ATP release and gene expression indicates that the Ca^{2+} current through *Cav1.1* is not relevant to activate the *E–T* coupling process. More importantly, it suggests the existence of a Ca^{2+} -current-independent function of *Cav1.1*, related to activation of *E–T* coupling that is blocked by these two drugs. In fact, the voltage gated Ca^{2+} channel (driving an L-type current) function of *Cav1.1* is also irrelevant for the voltage sensor function in *E–C* coupling activation. Hence, the roles of *Cav1.1* as voltage sensor in *E–C* coupling and *E–T* signaling, both activated by membrane depolarization, are independent of the function of *Cav1.1* as Ca^{2+} -current carrier.”

Since the topic of our study is exclusively to investigate the relevance of skeletal muscle DHPR in its function as Ca^{2+} -conducting channel for muscle performance, we hope we could make clear why we can't address topics of muscle plasticity in our manuscript without highly confusing the readers since the underlying mechanisms are fully independent of DHPR Ca^{2+} influx.

In addition, the EK mouse model that lacks *CaV1.1* Ca^{2+} influx does exhibit significant effects on muscle size, fiber type, protein metabolism, SR Ca^{2+} stores, fatigue, and age-related strength mediated in large part via *CaMKII* signaling (Lee et al. *Skeletal Muscle* 5:4 DOI 10.1186/s13395-014-0027-1, 2015). Although the authors address some of these findings in their Discussion, the current study would be strengthened if they address in more detail basic skeletal muscle phenotype (i.e., fiber type) and signaling pathways known to influence to phenotype. Moreover, the authors should consider adding the potential role of Ca^{2+} flux through the DHPR in affecting skeletal muscle phenotype in their Introduction (51-54).

In the Introduction section of the revised manuscript we mention (as also requested by reviewer #3) the *E11K* mouse model (Lee et al., *Skelet. Muscle*, 2015) with its phenotypic differences to wild-type – which is in strong contrast to our study (lines 60-68). Due to space limitations, we give only a short view on the putative reasons for all these discrepancies (viz. the severe leakiness for monovalent cations in the *E11K*, but not in the *ncDHPR* model), and discuss all these aspects extensively in the Discussion section (lines 377-403). Additionally, in the Introduction section we now mention a putative role of the DHPR Ca^{2+} influx in the regulation of acetylcholine receptor pre-patterning, as well as its hypothetical negative inotropic effect, as postulated by Menezes-Rodrigues, F. S. *et al.*, (*Eur. J. Pharmacol.*, 2013), as a crucial mechanism in overall muscle health and fibre integrity (lines 53-56). Again both aspects are considered to more detail in the Discussion section (lines 325-330 and 353-376).

We agree with the reviewer that showing fibre-type composition – as an element of the basic skeletal muscle phenotype – strengthens our study, irrespective of the fact that we did not find any muscle phenotype in any of our experiments at *ex vivo* and whole animal levels. Therefore, and also in accordance to the wish of reviewer #1, we implemented fibre-type analysis in our revised manuscript. Results are added in Supplementary section as Fig. S6 and in the Result section: lines

139-144. In addition, the following note was added to the Discussion section (lines 331-333): “.....
*lack of DHPR Ca²⁺ influx in ncDHPR mice is inconsequential for body weight development, fertility,
locomotor activity, muscle fibre-type composition, fibre CSA, muscle strength and susceptibility to
fatigue”*

The battery of functional tests is impressive, however the authors should provide references that indicate that they are accurate and reliable measures of what they report to measure, especially given the extremely low statistical power values (e.g., 0.05 and 0.13 for mean latency to fall during 600-s Rotarod endurance task at constant speed) on a number of the tests.

Wherever possible we provide more references to indicate the accuracy and reliability of our functional tests (revised manuscript; lines 147, 151, 183, 185, 192, 203 and 215).

In addition, critical measures of isolated muscle function (i.e., twitch and tetanic forces) appear to have been made at relatively low temperatures based on the shape of the force-frequency relationship. Please provide the temperatures at which the physiological experiments were conducted (i.e., Methods or Figure legends), and if the measurements were made at relatively low temperatures, the authors should provide a rationale and address any potential impact on their conclusions.

To keep our twitch- and tetanic force measurements comparable to a published key study in this aspect (Ursu et al., *J. Physiol.*, (2001)), and also to the papers of Westerblad et al., *Am. J. Physiol.*, (1991); Johansson et al., *Am. J. Physiol.*, (2000); Johansson et al., *J. Physiol.*, (2003); Payne et al., *Exp. Gerontol.*, (2009) - and particularly to the paper of Menezes-Rodrigues et al., *Eur. J. Pharmacol.*, 2013, (Ref. 29) where we strongly contradict the author's conclusions - we performed all our recordings at room temperature (~26 °C). In the revised manuscript this fact is now indicated in the Methods section (lines 474 and 569) as well as in the Figure legends of Fig. 6, Supplementary Figure S7, Supplementary Figure S8, and Supplementary Figure S13. Since we routinely recorded wt and ncDHPR mice under identical conditions we don't expect any potential impact of this temperature aspect on our conclusion that there is no difference between both genotypes regarding skeletal muscle contraction forces.

Considering that 10 out of 12 genes of interest in the ncDHPR mice exhibited an approximate 10-20% lower expression than WT mice, the study would be greatly enhanced if the corresponding protein levels were also determined in view of their direct role in EC coupling and potential to influence cell signaling.

As suggested by the reviewer (as well as reviewer #1) we performed quantitative western blot analysis of key triadic EC coupling proteins with the aim to evaluate putative changes also at the protein level. Results are added in the revised manuscript as Fig. 9b, Supplementary Fig. S11 and S12 and in the Result section; lines 252-258, as well as briefly summed up in the Discussion section (lines 338-340).

Suggested minor comments to enhance readability of paper:

(49-51) The following sentence is somewhat awkward and should be revised for clarity: Even though, the physiological role of this enigmatic Ca²⁺ influx which is not (immediately) required for EC coupling is under debate since nearly half a century, a convincing answer elucidating its function is still in the dark.

As suggested by the reviewer, we revised the sentence to the following statement: *However, the physiological significance of this DHPR Ca²⁺ current, which is certainly not (immediately) required for EC coupling, is still unresolved despite being under investigation since nearly half a century*” (revised manuscript; lines 49-51).

Please provide the actual p values instead of directional p values (i.e., p>0.05). There are several cases where genotype differences are relatively large, e.g., line 174, ncDHPR exhibited 53% greater wire hang strength than WT (p value is 0.071 with a power of 0.32). Therefore, with a low statistical power (accepted values are typically 0.80) and a p value close to 0.05, drawing a definitive conclusion that there is no difference with this result is tenuous.

We would prefer to keep the directional *P*-values in our paper because these are most frequently used in the most renowned journals in our field and rather seem to be standard in the biological and biomedical papers published in *Nature Communications*. We think that directional *P*-values are immediately recognized by the reader and thus allows more fluent reading compared to the use of individualized *P*-values.

Regarding the incriminated data set (Fig. 7a) on wire-hang tests with our aged mice (17-22 months old; average age: 19.3 months), the reviewer is right that the mean value of the aged ncDHPR mice is 53% higher than wt and thus the *P*-value is statistically insignificant with “only” 0.07.

Tested aged individuals, due to the different manifestation of the aging process, show a much higher individual variability in physiological or pharmacological tests - compared to an inherently physiologically more homogenous young test population (Rietbrock, N., *Clinical Pharmacology in the Aged. Methods Clinical Pharmacology*, 1985). Especially, if a test shows such extreme age dependence like in the case of the wire-hang test (Fig. 7a – compared to the less age-dependent grip-strength test in Fig. 7b using the same cohort of test animals), a couple of very fit individuals can produce a strong positive shift in the mean value and a decline in the *P*-value. With this in mind, we tried to do the best for the quality of our experiment and used a high number of aged mice (25 + 27

individuals!) for all tests. However, in the case of the ncDHPR hang-test series two aged individuals showed *Latency-to-fall*-values comparable to the young group and thus pulled the mean value up and brought the *P*-value down. To make this data set more transparent, we show here a *box plot* of the hang test on aged mice. This *box plot* shows the two outliers of the ncDHPR group. A recalculation of our data set without the two outliers lead to the following values: aged ncDHPR (123.7±15.4 s; n=23) and wt (100.44±15.06 s; n=27) with a *P*-value (Student’s *t*-test) of 0.29. With no rational to

remove these two outliers from our original data set, we presented the data as they are.

(106-108) Please indicate that voltage-induced SR Ca²⁺ transients were also normal in the following sentence: Altogether, these results indicate a complete elimination of skeletal muscle Ca²⁺ influx in the ncDHPR mouse model and thus serves as a unique system for investigating the physiological relevance of the DHPR Ca²⁺ influx in skeletal muscle EC coupling.

Based on the reviewer's suggestion, we changed the sentence as follows: "*Altogether, these results indicate a complete elimination of skeletal muscle Ca²⁺ influx but unaltered voltage-induced SR Ca²⁺ release in the ncDHPR mouse model. Thus, the ncDHPR model serves as a unique system for investigating the physiological relevance of the DHPR Ca²⁺ influx in skeletal muscle EC coupling.*" (revised manuscript; lines 117-120).

(107-125) The use of the word "initial" seems odd given that the data describes adult animals, and the word "phenotype" leaves the reader expecting more information about the mouse other than just its body weight and litter size. Please consider revising and combining this data with measurements of skeletal muscles described later and ideally fiber type composition.

As suggested by the reviewer, in the revised version of our manuscript this paragraph is now titled "*Basic phenotypical characterization....*" instead of "*Initial phenotypical characterization....*" (line 129). In agreement with the advice of the reviewer, we also added weight, length, and diameter measurements of SOL and EDL muscles, as well as fiber-type composition (lines 134-144).

(126-128) Suggest revising to read "Since the ncDHPR mouse did not show any evident phenotype differences from wt,..."

As suggested by the reviewer, the sentence was revised (revised manuscript; lines 145-146).

(128) Suggest revising to read "Home-cage ambulatory activity recorded..."

Also this sentence was revised and the term "*ambulatory*" has been added (revised manuscript; line 147-148, line 480 and legend to Fig. 5).

(132-133) Please add the average age of the mice.

The average age of the mice has been added (revised manuscript; lines 149, 151, and 159).

(134, 150) Replace "identical" with "similar" (the values are not identical).

The term "*identical*" was replaced by "*similar*" (revised manuscript; lines 153 and 166).

(140) Mouse soleus muscle is more "slow-twitch" than mouse EDL, however it is not predominately "slow" from a traditional fiber-type classification based on MHC isoforms (e.g., Pelligrino et al Eur J Appl Physiol. 2005 Mar;93(5-6):655-64).

Based on the reviewer's advice we tried to describe the "partially slow-twitch character" of mouse soleus muscle more accurately by stating: "*To identify possible muscle-type specific differences we chose extensor digitorum longus (EDL) muscle which primarily consists of fast-twitch fibres and soleus (SOL) muscle, which is more a slow-twitch muscle compared to EDL*" in the revised manuscript (lines 134-136).

(142) Please provide the average age of the "aged" mice (14 months of age is not considered old for these mice).

As suggested, we added the average age of the old, but also of the young mice (lines 149, 151 and 159) at this position. In this context we apologize for a typo that led to confusion and was corrected in the revised manuscript. Our "aged" mice are not between "14-22 months" of age as erroneously stated, but "17-22 months" (revised manuscript; lines 159-160). We thank the reviewer for pointing out this typo!

(174) This functional test is a measure of muscular endurance and not maximal muscular strength.

According to Brooks & Dunnett, *Nat. Ref. Neurosci.* (2009); Deakon, *J. Vis. Exp.* (2013); Jansone et al., *PLAS*, (2016) the classical "wire hang test" or "inverted screen test" can be considered as an accurate functional test for muscle strength using all four limbs (beside balance, coordination, and muscle condition / endurance). The same opinion was shared by the "Treat-NMD, Neuromuscular Network" of the Wellstone Muscular Dystrophy Center, Washington DC (2016) (http://www.treat-nmd.eu/downloads/file/sops/dmd/MDX/DMD_M.2.1.004.pdf) and (http://www.treat-nmd.eu/downloads/file/sops/dmd/MDX/DMD_M.2.1.005.pdf). Therefore we did not change the description of the hang-test outcome in the specific Results paragraph "*ncDHPR mice display normal muscle coordination and strength*", however removed the expression "*maximal*" (revised manuscript; line 185). In addition, we included the citation (Ref. 37) of Brooks & Dunnett, *Nat. Ref. Neurosci.* (2009) to indicate that the hang test is also an accurate and reliable measure of muscle strength (lines 183 and 185).

(190-191) Please revise the following "in the absence of extracellular Ca²⁺ influx in ncDHPR mice" because the authors did not measure total Ca²⁺ flux into muscle fibers nor through any other Ca²⁺ channels (e.g., SOCE, SAC).

We fully agree with the reviewer and the sentence has been corrected. In the revised version it reads as follows: "*... in the absence of DHPR-mediated Ca²⁺ influx in ncDHPR mice.*" (revised manuscript; lines 199-200).

(229-230) The authors state that "...either influx or efflux of Ca²⁺ ions compensates for the reduced myoplasmic Ca²⁺ concentration in ncDHPR mice." However, the authors should revise this statement since they only measured Ca²⁺ flux through ncDHPR and not through other Ca²⁺ channels or the [Ca²⁺]_i after

contractile activity.

We agree with the reviewer and corrected this misleading statement in the revised manuscript. The revised sentence reads as follows: “.....*whether up- or down-regulation of any of the key triadic proteins responsible for either influx or efflux of Ca²⁺ ions compensates for the lack of DHPR-mediated Ca²⁺ influx in the ncDHPR mice.*” (revised manuscript; line 233).

(268) Please add “force” in the following: ...robust increment in muscle twitch contraction force in wt mice.

The term “*force*” has been inserted in the revised manuscript, (line 277).

Our answers to the comments of **Reviewer #3.**

This is a clearly developed and articulated study that provides comprehensive and clear-cut tests with respect to the somewhat controversial functional role of Ca influx via the L-type Ca channel (LTCC) in mammalian skeletal muscle. The authors have used a knock-in mouse that expresses an LTCC mutant that completely prevents current via the Ca channel. This mutation also prevents monovalent cation current via the LTCC, which is a potential limitation for an excellent 2015 knock-in mouse study of similar overall strategy (ref 53). In my mind this manuscript provides definitive evidence that the residual Ca current that is carried by the normal mouse skeletal muscle LTCC (Cav1.1) is vestigial, and that its loss has no discernable functional consequence. The work is comprehensive from biophysical characterizations to integrated function, behavior and aging. This study is a major advance for this field. My only comments are for minor stylistic and clarifying points.

1) The related 2015 paper from Hamilton’s group should probably be mentioned in the Introduction section (not just at the end of the Discussion). That prior study used the same “overall” strategy, and the functional distinction between the ‘EK-mutant’ and the N617D mutant used here is important in both justifying the novelty of this study, and the implications of the results. That is, it can set up the present study in an even stronger way, and I would say that this could be done without lengthening the Introduction.

We agree with the reviewer that mentioning the publication of Lee et al., 2015 already in the Introduction section can present our study in an even stronger way. Accordingly, we introduced the EK-mutant of the Hamilton lab and its functional distinction to our N617D mutation, in the Introduction section of the revised manuscript, (lines 60-68).

2) Related to that manuscript length point, the overall style is somewhat more wordy than necessary. The Intro could be a bit more concise, without loss of content and the Results which repeat nearly every number that is already displayed in the figures (and sometimes to 5 significant(?) figures). I suggest that reciting the numbers be reserved for a few key points that the authors want to make (even if they prefer to put the actual values in the legends. I do understand their desire to be comprehensive in that way, but it does break up the flow of the text.

We tried to make the Introduction less wordy by deleting some less important sentences. The *mean ± standard error* values are to a big extent removed and shifted from the Results section to the Figure legends and as suggested by the reviewer only key points were left in the main text.

Other minor style points that made reading of this otherwise very clear manuscript more difficult as a reviewer was the lack of clear paragraph breaks (by indents or extra spacing) and lack of numbers on the Figure pages. Obviously these are issues the copy editing will remedy, so this is mainly advice for future.

We apologize for the sparse use of paragraph breaks and for not numbering the figure pages. In our revised manuscript, we introduced more paragraph breaks and numbers on the figure pages.

3) On line 229 it is implied that there is reduced myoplasmic Ca in the ncDHPR mice, but no differences in Ca transients, Ca content or resting [Ca]_i measurements were reported. I think you mean "...for the reduced Ca influx via DHPR in the ncDHPR mice." or "...for the possible reduced myoplasmic Ca levels that might have been anticipated in the ncDHPR mice."

We thank the reviewer for pointing out this misleading statement. The corrected sentence reads as follows: "*.....whether up- or down-regulation of any of the key triadic proteins responsible for either influx or efflux of Ca²⁺ ions compensates for the lack of DHPR-mediated Ca²⁺ influx in the ncDHPR mice.*" (revised manuscript; line 233).

In this same spot (In 230 & Fig 9), the graphs should indicate that these were mRNA measurements, so that one immediately knows upon looking at the Fig that this is message, not protein. Likewise on line 239, change to ""Furthermore, expression profiles of mRNA that codes for critical Ca...".

We agree with the reviewer and in the revised manuscript, graphs in Fig. 9 and Supplementary Fig. S10 indicate that these were mRNA measurements. We added the term "*mRNA level*" to the figures at accurate positions. In the text the term "*.... expression profiles of mRNAs that code for*" was added (revised manuscript; lines 243-244).

Minor edits:

Line 270; replace "as well" with "also"

"as well" was replaced by *"also"* (revised manuscript; line 279).

Line 276; Waning of the response was not apparent during the 25 min exposure period, so I suggest deleting "was transient and the effect".

"was transient and the effect" was deleted (revised manuscript; line 283).

Line 289; replace "recent function" with "functional importance"

"recent function" was replaced by *"functional importance"* (revised manuscript; line 296).

Line 307; delete “this mysterious”

“this mysterious” was deleted (revised manuscript; line 315).

REVIEWERS' COMMENTS:

Reviewer #1 (Remarks to the Author):

An outstanding manuscript. No further comments.

Reviewer #2 (Remarks to the Author):

The authors have adequately attended to my previous comments and are commended on their comprehensive analysis of this intriguing research question.

Minor correction: Change "Bras" to Bars in last line of the legend for Supplementary Figure S5. Physical muscle parameters are unaltered in ncDHPR mice.

Reviewer #3 (Remarks to the Author):

The authors have been highly responsive and have improved the manuscript.

Response to the referees:

Again we want to thank all three referees for their overall constructive and valuable input that considerably improved our manuscript and made our study more “air-tight”. Our answers are in blue.

REVIEWERS' COMMENTS:

Reviewer #1 (Remarks to the Author):

An outstanding manuscript. No further comments.

Thank you!

Reviewer #2 (Remarks to the Author):

The authors have adequately attended to my previous comments and are commended on their comprehensive analysis of this intriguing research question.

Minor correction: Change “Bras” to Bars in last line of the legend for Supplementary Figure S5. Physical muscle parameters are unaltered in ncDHPR mice.

The typo in Supplementary Figure 5 was corrected.

Reviewer #3 (Remarks to the Author):

The authors have been highly responsive and have improved the manuscript.

Thank you!